# GORAB scaffolds COPI at the *trans*-Golgi for efficient enzyme recycling and correct protein glycosylation

Tomasz M. Witkos[1], Wing Lee Chan[2,3], Merja Joensuu[4,10,11], Manuel Rhiel[5], Ed Pallister[6], Jane Thomas-Oates [6], A. Paul Mould[1], Alex A. Mironov[1], Christophe Biot [7], Yann Guerardel [7], Willy Morelle[7], Daniel Ungar [8], Felix T. Wieland[5], Eija Jokitalo[4], May Tassabehji[1,9], Uwe Kornak[2,3] & Martin Lowe[1]

COPI is a key mediator of protein trafficking within the secretory pathway. COPI is recruited to the membrane primarily through binding to Arf GTPases, upon which it undergoes assembly to form coated transport intermediates responsible for trafficking numerous proteins, including Golgi-resident enzymes. Here, we identify GORAB, the protein mutated in the skin and bone disorder gerodermia osteodysplastica, as a component of the COPI machinery. GORAB forms stable domains at the *trans*-Golgi that, via interactions with the COPI-binding protein Scyl1, promote COPI recruitment to these domains. Pathogenic GORAB mutations perturb Scyl1 binding or GORAB assembly into domains, indicating the importance of these interactions. Loss of GORAB causes impairment of COPI-mediated retrieval of *trans*-Golgi enzymes, resulting in a deficit in glycosylation of secretory cargo proteins. Our results therefore identify GORAB as a COPI scaffolding factor, and support the view that defective protein glycosylation is a major disease mechanism in gerodermia osteodysplastica.

[1] School of Biology, Faculty of Biology, Medicine and Health, University of Manchester, The Michael Smith Building, Oxford Road, Manchester M13 9PT, UK. [2] Berlin-Brandenburg Centre for Regenerative Therapies (BCRT), Institut fuer Medizinische Genetik und Humangenetik, Charité – Universitätsmedizin Berlin, corporate member of Freie Universität Berlin, Humboldt-Universität zu Berlin and Berlin Institute of Health, Berlin 13353, Germany. [3] FG Development & Disease, Max Planck Institut fuer Molekulare Genetik, Berlin 14195, Germany. [4] Cell and Molecular Biology Program, Institute of Biotechnology, University of Helsinki, Helsinki 00014, Finland. [5] Heidelberg University Biochemistry Center, Heidelberg University, Heidelberg 69120, Germany. [6] Department of Chemistry, University of York, York YO10 5DG, UK. [7] Univ. Lille, CNRS, UMR 8576 - UGSF - Unité de Glycobiologie Structurale et Fonctionnelle, F-59000 Lille, France. [8] Department of Biology, University of York, York YO10 5DD, UK. [9] Manchester Centre for Genomic Medicine, St. Mary's Hospital, Manchester Academic Health Sciences Centre (MAHSC), Manchester M13 9WL, UK. [10] Present address: Clem Jones Centre of Ageing Dementia Research, Queensland Brain Institute, The University of Queensland, Brisbane, Brisbane QLD 4072, Australia. [11] Present address: Minerva Foundation Institute for Medical Research, 00290 Helsinki, Finland. Correspondence and requests for materials should be addressed to M.L. (email: martin.lowe@manchester.ac.uk)

COPI (coat protein complex I)-coated transport vesicles mediate protein trafficking in the early secretory pathway. They are responsible for retrograde transport from the Golgi apparatus to the endoplasmic reticulum (ER)[1], and for trafficking between cisternae within the Golgi apparatus[2–4]. Within the Golgi, COPI-coated vesicles mediate retrograde traffic of Golgi resident enzymes[5,6], and may also participate in anterograde trafficking of certain cargoes[7,8]. Although COPI is best known for its role in vesicle trafficking, recent studies also suggest possible involvement in trafficking via tubular intermediates at the level of the Golgi stack[9]. In line with its trafficking functions, COPI is localized at the ER-to-Golgi intermediate compartment (ERGIC) and Golgi apparatus, where it is abundant at the cisternal rims and enriched towards the cis-side[10,11]. The COPI coat is comprised of the hetero-heptameric coatomer complex[2], which is recruited from the cytosol to the membrane by the small GTPase Arf1[12,13], which itself is recruited from the cytosol concomitant with guanosine 5'-triphosphate (GTP) loading[14]. Coatomer functions to both select cargo and promote vesicle formation[15,16], which is facilitated by the assembly of coatomer complexes into a cage-like structure[17,18]. Although Arf1 is the primary driver of coatomer recruitment, additional factors may contribute to this process. The p24 family proteins have been proposed to function as coatomer receptors[15,19], but the extent to which other proteins participate in coatomer recruitment or assembly is poorly understood.

The cutis laxa syndromes are defined by the presence of loose, wrinkly, inelastic skin and can be classified into various types depending upon clinical features and the gene that is mutated[20]. The skin phenotype seen in cutis laxa is thought to arise from defective production and/or assembly of extracellular matrix, predominantly at the level of elastic fibers[21]. Mutations in several elastic fiber proteins have been shown to cause cutis laxa but, interestingly, causative mutations in several cellular proteins have also been identified[20,21]. Among these is GORAB, also known as Scyl1BP1, whose mutation is responsible for gerodermia osteodysplastica (GO)[22]. The hallmark symptoms of GO are cutis laxa and osteoporosis, with reduced bone mass and susceptibility to fractures[23,24]. As both symptoms are features of aging, GO has been classified as a progeroid disorder[22]. Hence, understanding how loss of GORAB leads to pathological changes in skin and bone is likely to give new insight into how these tissues age.

GORAB is localized to the trans-side of the Golgi apparatus[22]. It is comprised of a central coiled-coil region that is responsible for Golgi targeting, most likely via interactions with the small GTPases Rab6 and Arf5[25]. It has been proposed that GORAB is a member of the golgin family of coiled-coil Golgi proteins[22], which participate in vesicle tethering[26,27]. GORAB has also been proposed to function as a transcriptional activator for neurite outgrowth[28], as a modulator of MDM2 ubiquitylation that in turn can impact upon p53 levels and apoptosis[29], and has recently been shown to play a role in centriole duplication and ciliogenesis[30,31]. Despite these advances, the function of Golgi-associated GORAB remains poorly defined, and the pathogenic mechanism underlying GO remains to be determined. Here, we show that GORAB functions in intra-Golgi trafficking as a scaffolding protein for COPI. It forms stable membrane domains that, via interaction with Scyl1, stabilize COPI assembly at the trans-Golgi. Loss of GORAB function results in reduced recycling of trans-Golgi enzymes and improper glycosylation of cargo proteins within both cultured cells and skin tissue. Our results therefore identify GORAB as a player in COPI trafficking, and provide a mechanism to explain the symptoms of GO that are also relevant to human aging.

## Results

**GORAB interacts with Scyl1.** To gain insight into the cellular functions of GORAB we first investigated its interaction partners. GORAB was first identified as a potential binding partner for Scyl1[32], also known as NTKL, but this interaction has yet to be validated. We therefore determined whether Scyl1 is a bona fide interactor of GORAB. GORAB bound to Scyl1 in the yeast two-hybrid system (Fig. 1a). GORAB and Scyl1 self-association was also detected in the yeast two-hybrid system, consistent with the presence of coiled-coil and HEAT repeat domains respectively, in these proteins (see Fig. 1d, e)[33]. GORAB and Scyl1 interaction was confirmed in protein pull-down experiments (Fig. 1b). The binding between GORAB and Scyl1 is direct, as indicated by pull-down experiments with purified recombinant proteins (Fig. 1c). We next mapped the interaction sites in GORAB and Scyl1. GORAB is comprised of a central coiled-coil region, with several predicted breaks within the coiled-coil (Fig. 1d, right), flanked by non-coiled N- and C-terminal domains. Pull-down experiments with purified proteins indicated that the N-terminal non-coil domain is sufficient to bind Scyl1 (Fig. 1d, left). Scyl1 is comprised of an N-terminal kinase-like domain that is predicted to be catalytically inactive, centrally located HEAT repeats and a C-terminal short coiled-coil domain followed by a dibasic binding motif for the coatomer complex of the COPI vesicle coat[34] (Fig. 1e, right). Mapping experiments indicated that the binding site for GORAB resides within the kinase-like domain of Scyl1 (Fig. 1e, left). Thus, GORAB and Scyl1 are bona fide binding partners that directly interact via their respective N-terminal domains.

**GORAB forms discrete domains at the trans-Golgi.** GORAB was previously localized to the trans-Golgi by immunofluorescence microscopy[22,25]. We were therefore unsurprised to find extensive co-localization of GORAB with the trans-Golgi marker TGN46 by immunofluorescence microscopy (Fig. 2a). However, interestingly, closer inspection revealed that, unlike TGN46, GORAB was not evenly distributed throughout the trans-Golgi but rather concentrated in discrete puncta (Fig. 2a). The puncta disappeared upon depletion of GORAB, and were also observed with over-expressed green fluorescent protein (GFP)-tagged GORAB, confirming specificity of the staining (Fig. 2b). The discrete nature of the GORAB puncta was further revealed by super-resolution (Fig. 2c) and immuno-electron microscopy of both HeLa cells and dermal fibroblasts (Fig. 2d, e). The GORAB puncta are enriched at the trans-side of the Golgi and found predominantly within the tubulo-vesicular trans-Golgi network, as well as occasionally at the rims of the trans-most Golgi cisternae (Fig. 2d, e).

Previous work has shown that Scyl1 distribution is biased towards the cis-Golgi, with a significant pool in the ERGIC[34]. Labeling for Scyl1 indicated its presence in numerous puncta within the Golgi region and in more peripheral ERGIC (Supplementary Figure 1A). As expected, many of the Golgi puncta overlap with the cis-Golgi marker GM130; however, there was also significant overlap of Scyl1 puncta with TGN46, indicating that a pool of Scyl1 also resides at the trans-Golgi (Supplementary Figure 1A). In agreement, we observed co-localization of Scyl1 puncta with GORAB (Supplementary Figure 1B, see also Fig. 2f). Although GORAB cannot bind directly to COPI (Supplementary Figure 1C), Scyl1 does, via its extreme C terminus (Supplementary Figures 1C and 1D)[34], which interacts with the γ-COP appendage domain (Supplementary Figure 1E)[33], and a second site within the β'-COP subunit[35]. We therefore investigated whether the GORAB and Scyl1 puncta also contained COPI. As shown in Fig. 2f, using super-resolution

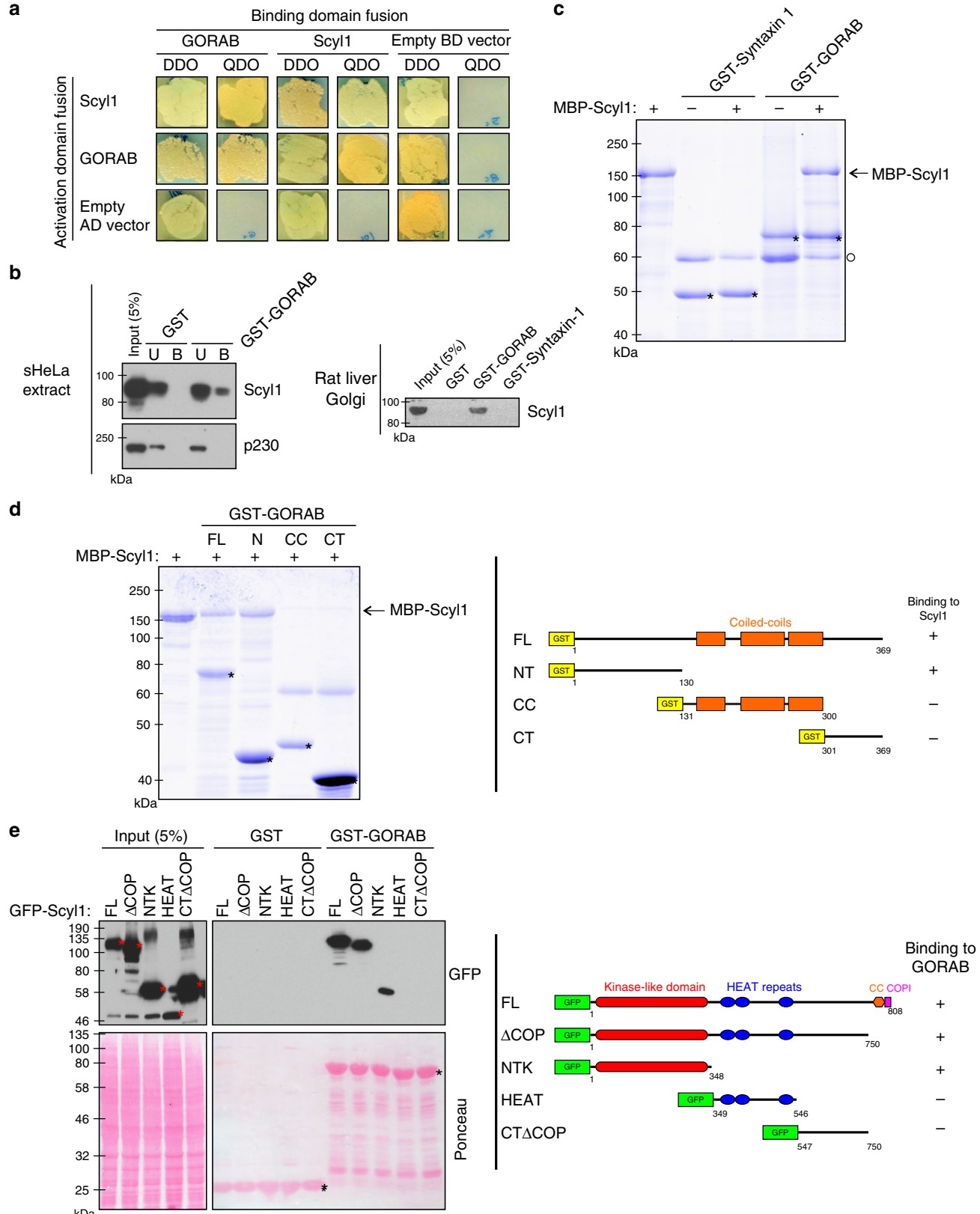

microscopy we could show that many of the GORAB- and Scyl1-positive puncta are also positive for COPI. Interestingly, Scyl1 frequently appeared to localize between GORAB and COPI, consistent with it bridging these two factors (Fig. 2f). We never observed COPI in the GORAB puncta in the absence of Scyl1, whereas the opposite could occur, consistent with the view that

Scyl1 is required for COPI association with the GORAB puncta (Fig. 2f). We could also detect overlap of Rab6 with the GORAB puncta, as expected from the known interaction of GORAB with Rab6[22], although Rab6 was also present outside these regions, consistent with it interacting with various effector proteins involved in different processes at the Golgi (Fig. 2g)[36].

**Fig. 1** GORAB interacts directly with the NTK domain of Scyl1. **a** Yeast two-hybrid assay between GORAB and Scyl1 constructs. Prey (AD-GORAB or AD-Scyl1) and bait (BD-Scyl1 or BD-GORAB) constructs were co-transformed into yeast and grown on double-drop-out (DDO) medium to verify expression of both proteins in transformants and on selective quadruple drop-out (QDO) medium to examine protein–protein interactions. Constructs containing AD and BD only were used as negative controls. **b** Pull-down assays with recombinant GORAB. Top panel: pull-down using bacterially expressed GST and GST-GORAB as bait and sHeLa cell lysate. Bottom panel: pull-down using GST, GST-tagged GORAB or Syntaxin 1 as bait and rat liver Golgi (RLG) membrane extract. Samples were blotted with the indicated antibodies. I input (5%), U unbound fraction (5%), B bound fraction (50%). **c** Pull-down assay using purified GST-Syntaxin 1 or GST-GORAB as bait and MBP-tagged Scyl1. Samples were subjected to SDS-PAGE and analyzed by Coomassie Blue staining. GST-tagged proteins are marked with an asterisk. BSA used as a carrier protein is marked with a circle. The faint bands running under MBP-Scyl1 correspond to likely degradation products or bacterial contaminants. **d** Mapping the binding site for Scyl1 on GORAB. Left, pull-down assay using purified GST-tagged GORAB fragments as bait and MBP-tagged Scyl1. Samples were subjected to SDS-PAGE and Coomassie Blue staining. GST-tagged proteins are marked with an asterisk. Right, schematic diagram of GST-tagged GORAB truncation constructs. **e** Mapping the interaction site for GORAB on Scyl1. Left, cell lysates obtained from cells transiently expressing GFP-tagged Scyl1 constructs were used for a pull-down assay with GST or GST-GORAB as bait and analyzed by western blotting. GST-tagged bait proteins and GFP-tagged proteins in inputs are marked with black or red asterisks respectively. Right, a schematic diagram of GFP-tagged Scyl1 truncation constructs. CC coiled-coil region, COPI COPI binding motif

**GORAB and Scyl1 are Arf effector proteins**. A recent study described binding of GORAB to Arf5, a Golgi-localized class II Arf[25]. Given the association of GORAB, via Scyl1, with COPI, and the fact that class I and II Arf GTPases both promote membrane recruitment of COPI[37], we re-evaluated GORAB interaction with Arfs. Using pull-downs, we could show that GORAB is able to bind to the class I Arfs, Arf1 and Arf3, in addition to Arf5 (Supplementary Figure 1F). Binding occurred only to the active, GTP-bound form and appeared strongest to class I Arfs. We also investigated Scyl1 binding to Arfs. It has been reported that Scyl1 binds selectively to class II Arfs, and that binding is independent of nucleotide status[33]. We observed binding of Scyl1 to Arfs, but binding was to class I Arfs only, with strongest binding to Arf1, and binding was only to the active GTP-bound form (Supplementary Figure 1F). Binding of both GORAB and Scyl1 to Arf1 is direct (Supplementary Figure 1G). These results suggest that GORAB and Scyl1 function as Arf effector proteins. Interactions between GORAB, Scyl1, COPI and Arf1 were not mutually exclusive, as indicated by pull-down experiments, consistent with the proteins functioning together in a complex (Supplementary Figure 1H).

**GORAB domains are stable entities**. To better understand the nature of the GORAB puncta (membrane domains), we investigated their dynamics. GFP-tagged GORAB was stably expressed at low levels and fluorescence recovery after photobleaching (FRAP) was performed. As shown in Fig. 3a, recovery of GFP-GORAB fluorescence was slow when compared to the GFP-tagged Golgi enzyme GalNAc-T2. This result indicates that GORAB is stably associated with the domains, and therefore that the domains themselves are stable entities. In contrast, recovery of GFP-tagged Scyl1 in the Golgi region was much faster, indicating that Scyl1 can rapidly exchange with the membrane (Fig. 3a and Supplementary Figure 2). Co-expression with mApple-GORAB decreased the rate of exchange of GFP-Scyl1 with the membrane, in addition to increasing the immobile fraction (Supplementary Figure 2). However, the rate of GFP-Scyl1 exchange remained significantly faster than that of GORAB (see Fig. 3a), supporting the view that Scyl1 rapidly exchanges with stable GORAB domains. The GORAB domains persist upon depletion of Scyl1 (Fig. 3b) or treatment of cells with brefeldin A (BFA) to remove Golgi-associated ARF and COPI (Fig. 3c), indicating that the domains can form independently of Scyl1, Arf and COPI.

**GO disease mutations disrupt Scyl1 binding and GORAB domains**. A number of disease-causing mutations have been described in the GORAB sequence, including several missense mutations[22,38,39]. Two recently described mutations identified in a compound heterozygous GO patient (F8L and K190del) are of

particular interest considering that neither mutation affects gross folding of GORAB or its targeting to the Golgi apparatus (Gopal-Kothandapani et al., in preparation) (Fig. 4a). These mutations must therefore affect another aspect of GORAB function. Pull-down experiments indicated that the F8L mutation does not affect binding of GORAB to Arf1, Arf5 or Rab6, or the ability of GORAB to self-associate (Fig. 4b). It does, however, greatly diminish binding to Scyl1, consistent with its location in the N-terminal Scyl1-binding region of GORAB (Fig. 4b). Surface plasmon resonance with purified proteins indicated high affinity binding of wild-type GORAB to Scyl1 (Kd of 0.52 nM), with the F8L mutant demonstrating a complete loss of binding (Fig. 4c and Supplementary Figure 3A). The pathogenic effect of the F8L mutation indicates that the GORAB–Scyl1 interaction is physiologically important.

Like F8L, the K190del mutant can also bind to Arf1, Arf5 and Rab6, although in this case binding to the Arfs is enhanced compared to wild-type GORAB (Fig. 4b). Binding to Scyl1 is not markedly affected by the K190del mutation, as indicated by pull-down (Fig. 4b) and surface plasmon resonance (Fig. 4c), which gives an identical binding affinity to wild-type GORAB ( Kd = 0.52 nM) (Supplementary Figure 3B). However, strikingly, there is a complete loss of GORAB self-association in the K190del mutant, as indicated by pull-down (Fig. 4b). The loss of GORAB self-association with the pathogenic K190del variant indicates this property of the protein is of physiological importance. Expression of the F8L and K190del variants in cells indicated that while the F8L still localizes to discrete domains, the K190del is unable to do so, and is evenly distributed through the trans-Golgi (Fig. 4d). Hence, self-association of GORAB is required for the assembly of the GORAB puncta. Together, the results indicate that both Scyl1 binding and oligomerization for stable domain assembly are required for full functionality of GORAB in vivo.

**GORAB and Scyl1 cooperate for COPI binding at *trans*-Golgi**. The ability of GORAB to form stable domains that also contain Scyl1, which in turn can bind to COPI, led us to propose that GORAB forms a scaffold that promotes COPI assembly at the *trans*-Golgi. To test this hypothesis, wild-type GORAB or mutants deficient in Scyl1 binding (F8L) or oligomerization (K190del) were expressed in cells and the stability of COPI membrane association assessed by treating cells with BFA. As shown in Fig. 4e, f, overexpression of wild-type GFP-tagged GORAB stabilized COPI association with the Golgi membranes, indicated by the retention of COPI in the perinuclear region following 10 min of treatment with BFA. This is in contrast to control cells, where COPI was completely cytosolic at the same time point. The stabilization of COPI at the Golgi was lost with the F8L or K190del mutants, indicating that both Scyl1 binding

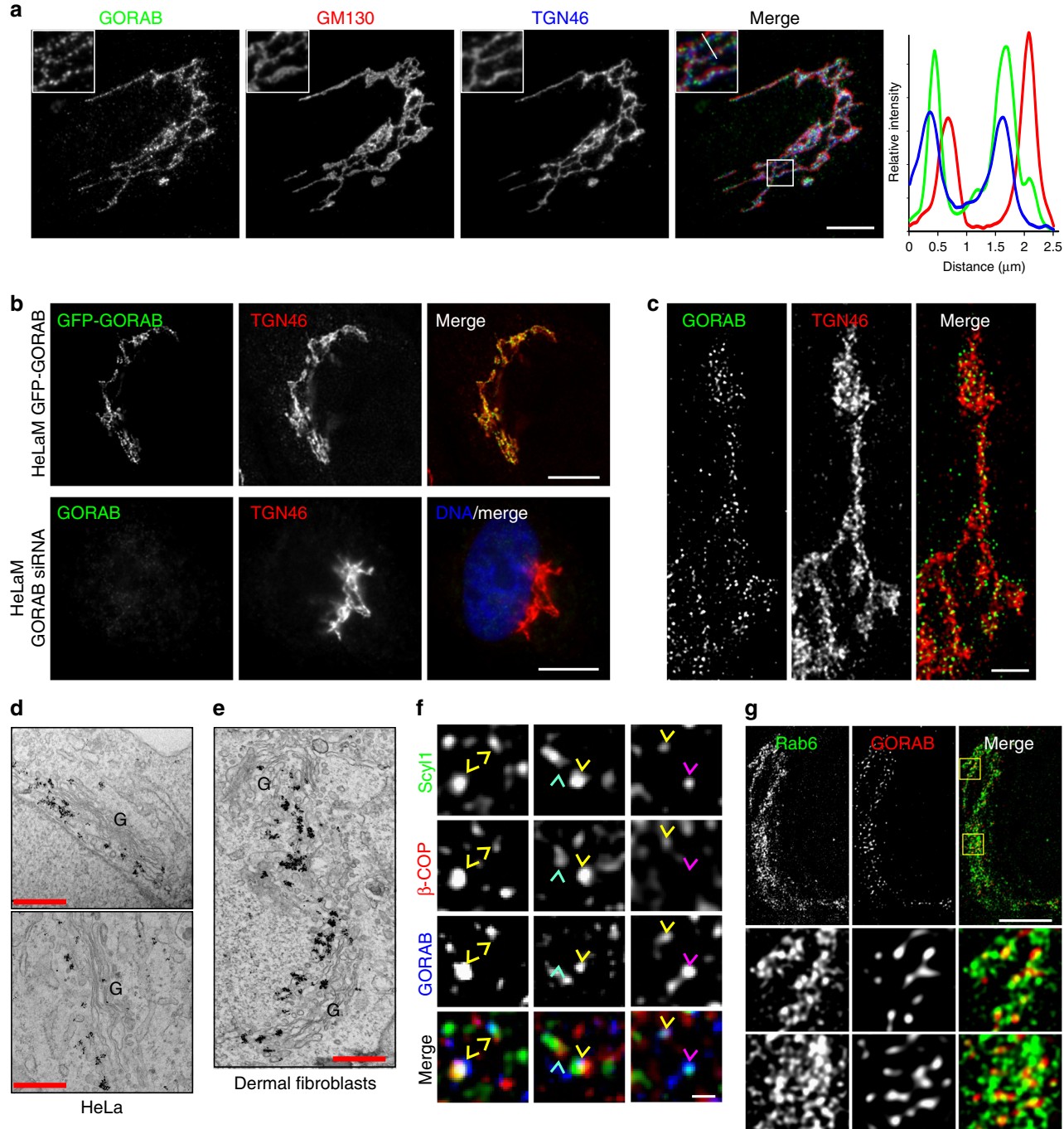

**Fig. 2** GORAB co-localizes with Scyl1 and COPI in discrete domains at the *trans*-Golgi. **a** Analysis of GORAB localization at the Golgi. Human dermal fibroblasts were fixed and labeled with antibodies to GORAB, GM130 and TGN46. Scale bar, 10 μm. The linescan is representative of data from *n* = 20 cells. **b** Analysis of GFP-GORAB localization in stably transfected HeLaM cells (top) and in HeLaM cells transfected with GORAB siRNA (bottom). Cells were fixed and labeled with antibodies to GORAB (bottom row only) and TGN46. Scale bar, 10 μm. **c** GORAB Golgi localization using STED microscopy. Human dermal fibroblasts were fixed and labeled with antibodies against GORAB and TGN46. Scale bar, 1 μm. **d**, **e** Representative EM micrographs depict localization of GORAB in HeLa cells (**d**) and human dermal fibroblasts (**e**). G Golgi. Scale bars, 500 nm. **f** Co-localization analysis of GORAB, Scyl1 and β'-COP using STED microscopy. Human dermal fibroblasts were fixed and labeled with antibodies against Scyl1, GORAB and β'-COP. Scale bar, 200 nm. Yellow arrowheads mark GORAB puncta co-localizing both with Scyl1 and β'-COP, magenta arrowheads mark GORAB puncta co-localizing with Scyl1 only and cyan arrowheads mark Scyl1 puncta co-localizing with β'-COP but devoid of GORAB. **g** Co-localization analysis of GORAB and Rab6 using STED microscopy. Human dermal fibroblasts were fixed and labeled with antibodies against GORAB and Rab6. Top, scale bar, 5 μm, bottom, scale bar, 200 nm

and oligomerization of GORAB are required to elicit this effect (Fig. 4f and Supplementary Figure 4A).

To further test the hypothesis, Scyl1 was also over-expressed in cells. The expression of Scyl1 stabilized membrane association of

COPI upon BFA treatment, which was evident both in puncta within the Golgi region that correspond to GORAB domains and in more peripheral puncta likely corresponding to the ERGIC (Fig. 4g–i). The ΔNTK Scyl1 mutant that cannot bind to GORAB,

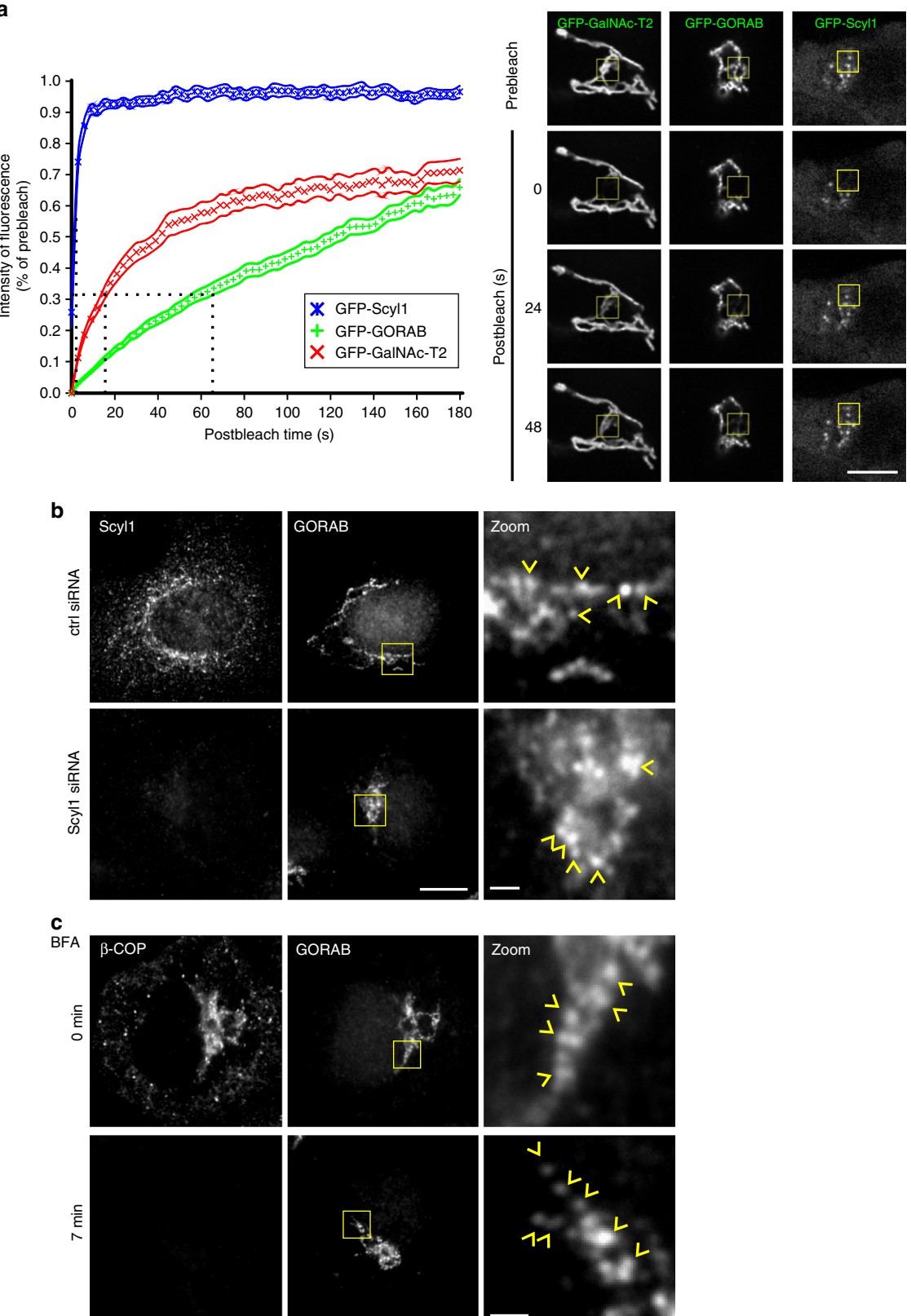

**Fig. 3** GORAB domains are stable entities. **a** Fluorescence recovery after photobleaching. Left, FRAP recovery curves for GFP-GalNAc-T2, GFP-GORAB and GFP-Scy1. Means with SEM for GFP-GalNAc-T2 ($n = 27$ cells), GFP-GORAB ($n = 28$ cells) and GFP-Scyl1 ($n = 18$ cells). Dotted lines mark points of half-time recoveries. Right, representative HeLa GFP-GalNAc-T2, HeLaM GFP-GORAB and HeLaM GFP-Scyl1 cells at pre-bleached and selected post-bleached states. Bleached region of interests are marked with yellow boxes. Scale bar, 10 μm. **b** Localization of GORAB in Scyl1-depleted cells. HeLa cells transfected with control or Scyl1 siRNA were fixed and labeled with antibodies to Scyl1 and GORAB. Scale bars, 10 μm and 1 μm. GORAB domains are marked with yellow arrowheads. **c** Localization of GORAB in BFA-treated cells. HeLa cells were exposed to 5 μg/mL BFA for 7 min prior to fixation and labeling with antibodies to Scyl1 and GORAB. Scale bars, 10 μm and 1 μm. GORAB domains are marked with yellow arrowheads

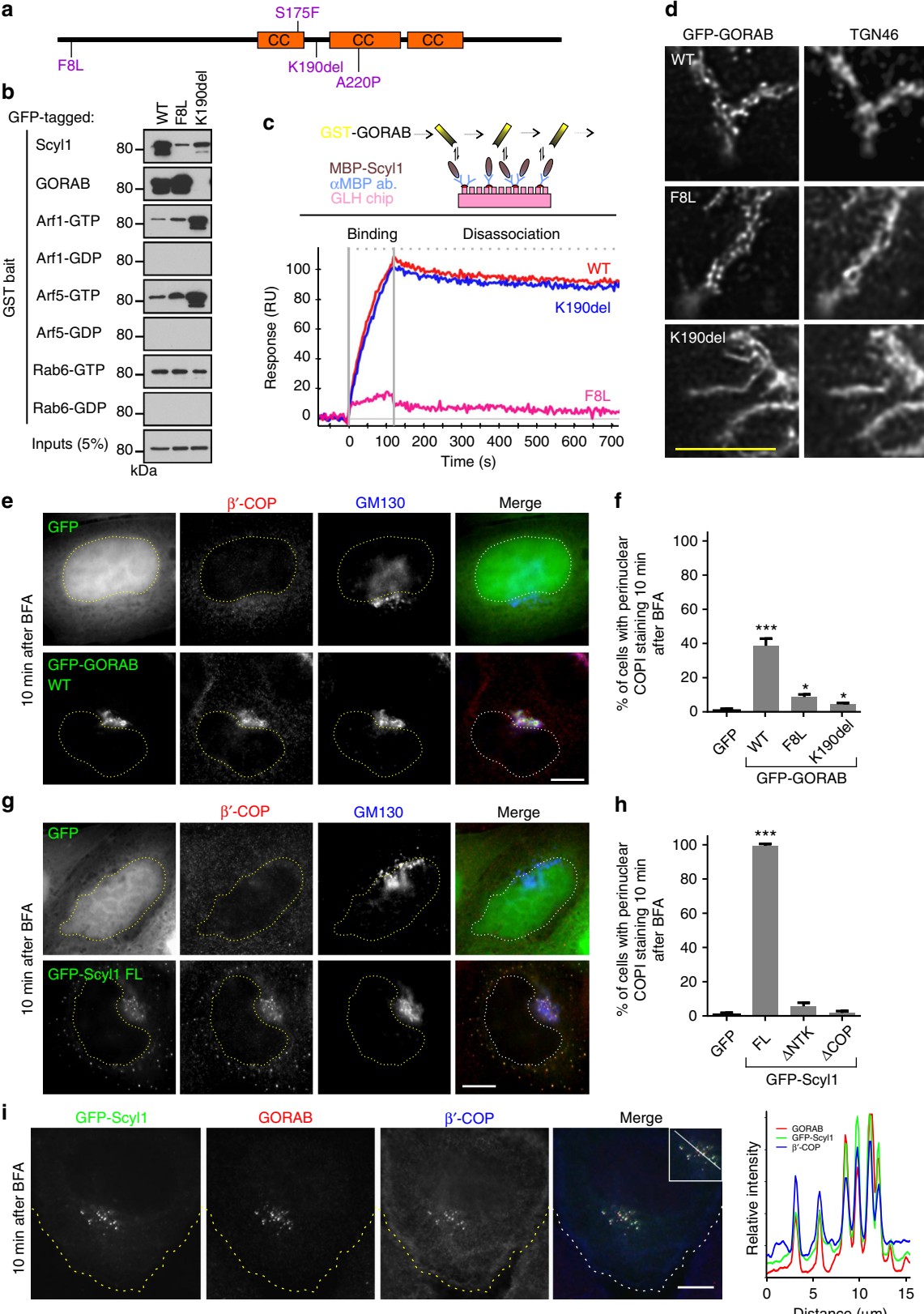

and the ΔCT mutant that cannot bind COPI, failed to stabilize COPI at the Golgi (Fig. 4h and Supplementary Figure 4B), indicating that Scyl1 must interact with GORAB and COPI to elicit this effect. Interestingly, the ΔNTK mutant was still able to stabilize COPI association with the ERGIC (Supplementary Figure 4B), indicating it can stabilize COPI at this compartment

independently of GORAB, consistent with the existence of at least two functionally distinct pools of Scyl1 in the secretory pathway. This view is further supported by the fact that Scyl1 recruitment to the Golgi, but not the ERGIC, requires binding to GORAB, as shown by the lack of Golgi localization of the ΔNTK mutant in untreated cells (Supplementary Figure 4C), and the loss of

**Fig. 4** Effect of pathogenic missense mutations upon GORAB behavior. **a** Location of known missense and single base deletion GORAB mutations in GO patients. Coiled-coil domains are depicted as orange rectangles. **b** Interaction of GORAB variants with GST-tagged bait proteins, as indicated, using cell lysates from RPE-1 cells expressing the indicated GFP-tagged GORAB variants. Inputs (5%) and bound fractions (50%) were blotted for GFP. **c** Surface plasmon resonance analysis of GORAB-Scyl1 binding. Top, experimental setup with a GLH sensor chip, cross-linked anti-MBP antibody, MBP-Scyl1 as bound ligand and GST-GORAB variants as analyte. Bottom, binding of GST-GORAB variants at 30 nM concentration for 120 s followed by 600 s disassociation. Similar results were obtained in three separate experiments. **d** Golgi localization of GFP-tagged GORAB variants using STED microscopy. RPE-1 cells were fixed and labeled with TGN46 antibodies. Scale bar, 5 μm. **e** COPI subcellular localization in HeLaM cells transfected with GFP or GFP-GORAB and incubated for 10 min with 5 μg/mL BFA. Cells were labeled with antibodies to β′-COP and GM130. Scale bar, 10 μm. Dotted line marks the nucleus. **f** Quantification of COPI retention in the Golgi region from **e**. Error bars represent mean ± SD, $n = 100$ cells in each of 3 independent experiments, $*p \leq 0.05$ and $***p < 0.001$, unpaired $t$-test. **g** COPI subcellular localization in HeLaM cells transfected with GFP or GFP-Scyl1 fixed 10 min after incubation with 5 μg/mL BFA. Cells were labeled with antibodies to β′-COP and GM130. Scale bar, 10 μm. Dotted line marks the nucleus. **h** Quantification of COPI retention in the Golgi region from **g**. Error bars represent mean ± SD, $n = 100$ cells in each of 3 independent experiments, $***p < 0.001$, unpaired $t$-test. **i** Co-localization between GFP-Scyl1, β′-COP and GORAB in HeLaM cells fixed 10 min after incubation with 5 μg/mL BFA. Cells were labeled with antibodies to GORAB and β′-COP. Scale bar, 10 μm. Dotted line marks the cell boundary. The white line indicates the pixels used for the RGB fluorescence intensity profile plot on the right, which is representative of data from $n = 20$ cells

Golgi-associated Scyl1 in GORAB-deficient fibroblasts (Supplementary Figure 4D). Over-expression of GORAB or Scyl1 had no effect upon membrane recruitment or BFA sensitivity of the *trans*-Golgi Arf-dependent clathrin adaptor complex AP1 (adaptor protein 1), as indicated by staining for γ-adaptin (Supplementary Figure 5A and B). Together, these results indicate that GORAB and Scyl1 associate to selectively stabilize recruitment of COPI at the *trans*-Golgi.

**GORAB and Scyl1 are sufficient for COPI membrane binding.** We next wanted to test whether GORAB and Scyl1 are sufficient to drive COPI membrane recruitment. For this purpose GORAB was relocated to mitochondria using a previously described inducible targeting method[40,41]. In this approach, GORAB containing a C-terminal FKBP tag was expressed in cells co-expressing mitochondrial targeted FRB, which binds to FKBP only in the presence of rapamycin, allowing inducible relocation of GORAB to mitochondria upon rapamycin addition (Fig. 5a). For these experiments we used the K190del mutant, which gave a clearer mitochondrial targeting, although similar results were obtained with wild-type GORAB. Cells were also treated with nocodazole to depolymerize microtubules and disperse the Golgi, giving a clearer readout[41]. In the absence of rapamycin, GORAB was localized to Golgi elements, where it co-localized with GFP-Scyl1, as expected (Fig. 5b). Upon addition of rapamycin, GORAB was efficiently relocated to mitochondria, and co-expressed GFP-Scyl1 (Fig. 5b), or endogenous Scyl1 (Fig. 5c), also redistributed to the GORAB-positive mitochondrial membrane. Golgi markers were absent from mitochondria under these conditions, excluding the possibility of gross distribution of Golgi elements (Supplementary Figure 6A). Endogenous COPI was partially localized to the GORAB and GFP-Scyl1 containing mitochondria, although there remained a significant amount in cytoplasmic puncta, likely corresponding to the ERGIC and dispersed Golgi elements (Fig. 5d). This result suggested that GORAB and Scyl1 can recruit COPI to mitochondria. To further assess this possibility, cells were treated with BFA to remove COPI from the Golgi and ERGIC. Under these conditions, there was almost complete redistribution of COPI to the mitochondria (Fig. 5d). Mitochondrial recruitment of COPI was not obvious in the absence of GFP-Scyl1 co-expression (Supplementary Figure 6B), likely due to the limiting amounts of endogenous Scyl1 in the cell compared to COPI[42]. Together, the results indicate that co-expressed GORAB and Scyl1 are sufficient to recruit COPI to the mitochondrial membrane. Moreover, it shows that COPI can be recruited to the GORAB-Scyl1 complex in the absence of membrane-associated Arf, which is further supported by the absence of mitochondrial Arf under conditions where COPI is

recruited there (Supplementary Figure 6C). As expected, Scyl1 deficient in GORAB (ΔNTK) binding failed to associate with mitochondria and recruit COPI, while the COPI binding mutant (ΔCT) was recruited to mitochondria but failed to recruit COPI (Supplementary Figure 7). Thus, GORAB recruits Scyl1, which in turn recruits COPI. In the same assay, we failed to observe mitochondrial relocation of AP1 (Supplementary Figure 8A). We also failed to observe GORAB-dependent mitochondrial recruitment of GFP-tagged Scyl2 or Scyl3, the latter of which has recently been proposed to function redundantly with Scyl1[43] (Supplementary Figure 8B). Lack of interaction between GORAB and Scyl3 was further confirmed in a pull-down experiment (Supplementary Figure 8C). Thus, GORAB selectively interacts with Scyl1, and the GORAB-Scyl1 complex is sufficient to drive selective membrane association of COPI (Supplementary Figure 8D).

Liposome binding experiments further supported a role for Scyl1 in promoting COPI association with membranes. As shown previously[19,44], incubation of synthetic liposomes with purified coatomer and Arf1 leads to recruitment of both proteins to the liposome membrane in a GTP-dependent manner (Fig. 6a). Scyl1 is also recruited to liposomes in the presence of Arf1 (Fig. 6a), consistent with its ability to bind directly to Arf1-GTP (Supplementary Figure 1G). When Scyl1 is added to liposomes in the presence of Arf1 and coatomer, Arf1 recruitment is not significantly altered, but recruitment of coatomer is increased nearly two-fold (Fig. 6b, c). Scyl1 is therefore able to enhance COPI recruitment to membranes in a manner independent of Arf1 association.

**Loss of GORAB causes defective protein glycosylation.** COPI is required for recycling of Golgi resident proteins, including the numerous enzymes that process glycans on cargo proteins and lipids as they transit the Golgi apparatus[45]. We therefore hypothesized that loss of GORAB would cause altered processing of cargo proteins due to impaired enzyme recycling. To test this possibility, dermal fibroblasts from wild-type or GO donors were subjected to *N*-glycomics analysis by mass spectrometry. This revealed a reduction in abundance of complex terminally sialylated glycans in the GO fibroblasts compared to wild-type controls, with a small reciprocal increase in their galactose terminated precursors, suggesting a deficit in addition of terminal sialic acid (*N*-acetylneuraminic acid (NeuAc)) residues in the GO cells (Fig. 7a). The deficit in terminal sialylation was confirmed using lectins. Immunofluorescence microscopy with fluorescently tagged *Maackia Amurensis* Lectin I (MALI) and *Sambucus Nigra* (SNA) lectins that specifically recognize sialic acid attached to terminal galactose or GalNAc via an α−2,3 linkage (MAL) or

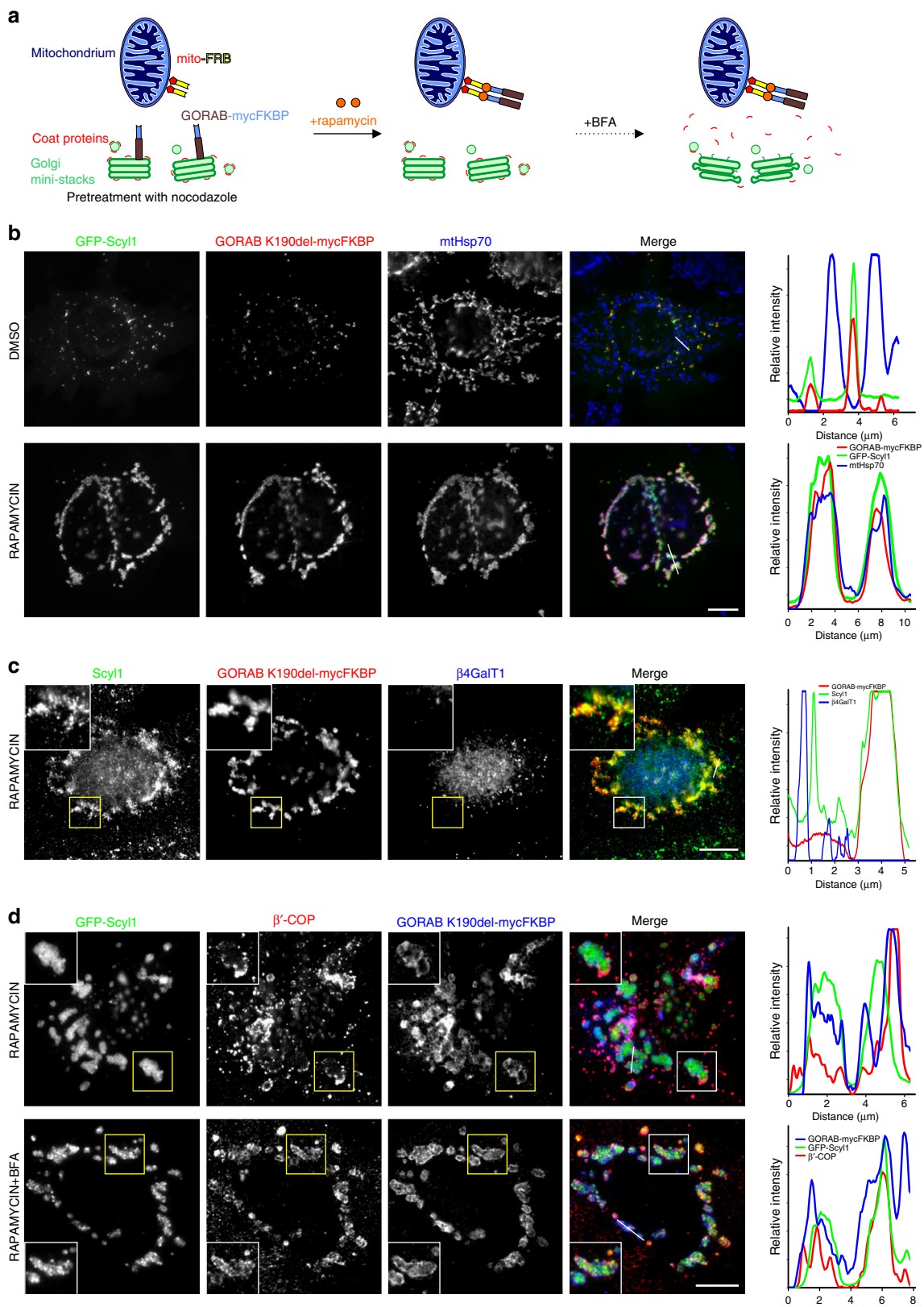

α−2,6 linkage (SNA), respectively, showed a significant reduction in SNA staining in GO fibroblasts compared to controls (Fig. 7b, c). In contrast, MAL staining was comparable between control and GO cells, indicating a preferential deficit in α−2,6 linkage of sialic acid to terminal galactose. Reduced SNA lectin staining was also evident by fluorescence-activated cell sorting analysis of GO

compared to control fibroblasts (Fig. 7d). To more directly assess glycosylation efficiency, cells were metabolically labeled with alkyne-tagged NeuAc precursor *N*-(4-pentynoyl) mannosamine (ManNAl), which allows fluorescence detection of sialic acid incorporation into glycoproteins within living cells[46]. GO cells incorporated less fluorescently tagged sialic acid at the

**Fig. 5** GORAB, via Scyl1, is sufficient to recruit COPI to membranes. **a** A schematic diagram depicting the mitochondrial relocation assay where the addition of rapamycin induces mitochondrial relocation of FKBP-tagged GORAB, allowing for recruitment of associated factors to this compartment. **b** Relocation of GORAB-mycFKBP and co-expressed GFP-Scyl1 to mitochondria. HeLaM cells co-transfected with mito-FRB and GORABK190del-mycFKBP constructs were pretreated with 2.5 µg/mL nocodazole for 2 h and further incubated with 1 µM rapamycin or DMSO for 3 h prior to fixation. Cells were labeled with antibodies to myc and mtHsp70. **c** Relocation of endogenous Scyl1 to mitochondria by GORAB-mycFKBP. HeLaM cells co-transfected with mito-FRB and GORAB-mycFKBP and treated as described in **b** and labeled with antibodies to endogenous Scyl1 and the Golgi marker β4GalT1. **d** Relocation of COPI to mitochondria by GORAB-mycFKBP. HeLaM cells co-transfected with mito-FRB, GORABK190del-mycFKBP and GFP-Scyl1 were treated as in **b**, and additionally with 5 µg/mL BFA for 15 min (lower panel only). Cells were labeled with antibodies to β'-COP and myc. In **b**–**d**, scale bars are 10 µm and white lines indicate the pixels used for the RGB fluorescence intensity profile plots shown on the right, which are representative of data from n = 20 cells

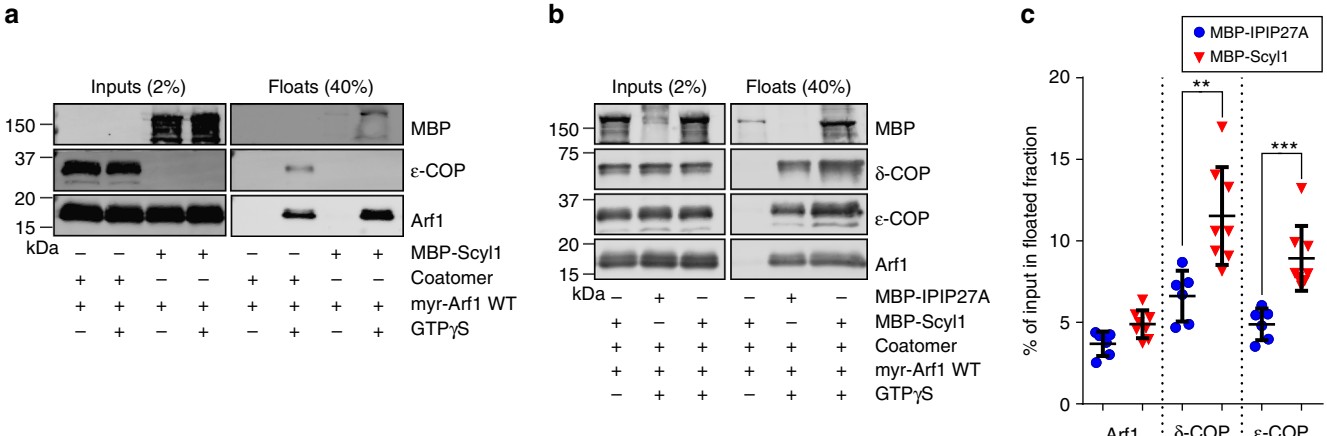

**Fig. 6** Scyl1-dependent recruitment of COPI to artificial membranes. **a** Liposome recruitment assay with purified MBP-Scyl1, myr-Arf1 and recombinant coatomer. Inputs (2%) and membrane-bound fractions (40%) were subjected to SDS-PAGE and blotted for MBP, ε-COP and Arf1. **b** Liposome recruitment assay with purified MBP-IPIP27A (as negative control), MBP-Scyl1 and myr-Arf1 and recombinant coatomer. Inputs (2%) and membrane-bound fractions (40%) were subjected to SDS-PAGE and blotted for MBP, δ-COP, ε-COP and Arf1. **c** Quantification of recruitment of Arf1 and coatomer (δ-COP, ε-COP) to liposomes in the presence of MBP-IPIP27A and MBP-Scyl1. Error bars represent mean ± SD, n = 6 independent experiments, **p < 0.01, ***p < 0.001, unpaired t-test

*trans*-Golgi compared to wild-type controls, indicating reduced sialylation upon loss of GORAB (Fig. 7e, f).

To assess whether loss of GORAB also caused altered glycosylation in vivo, skin samples were obtained from a GORAB-deficient knockout mouse[47] and analyzed by glycomics. The analysis revealed a striking reduction in complex *N*-glycans, which includes the species with terminal sialic acid residues (Fig. 7g), and a reciprocal increase in Mann5 oligomannose species (Fig. 7g). Blotting of mouse skin samples with lectins corroborated the mass spectrometry data, showing a reduction in high molecular weight species detected by the SNA and E-PHA lectins, which label α−2,6 linked terminal sialic acid and complex *N*-glycan chains respectively (Fig. 7h and i). Loss of GORAB therefore leads to perturbation of protein glycosylation in the Golgi apparatus, with a reduced abundance of complex and terminally modified glycoproteins. The phenotype is evident in vitro but appears to be more penetrant in vivo.

**Mislocalization of sialyltransferase upon loss of GORAB**. The glycomics and lectin data suggested a deficit in recycling of enzymes involved in generating complex terminally modified glycan species, most strikingly terminal α−2,6 sialylation, in GO cells. Defective recycling would be expected to cause a shift in enzyme distribution to later Golgi compartments, or even to post-Golgi compartments. Due to the lack of reagents to label endogenous ST6GAL1 and ST6GALII, we generated a HeLa cell line stably expressing horseradish peroxidase (HRP) fused to ST6GALI (designated ST-HRP). This fusion allows localization to be performed at high resolution using cytochemical staining followed by electron microscopy, and has previously been used to

track *trans*-Golgi morphology during the cell cycle[48]. In control cells treated with luciferase control small interfering RNA (siRNA) (Fig. 8a), ST-HRP was predominantly localized to the *trans*-most cisterna of the Golgi stack, with some additional signal present in adjacent tubulo-vesicular profiles corresponding to the *trans*-Golgi network (TGN) (Fig. 8b, c). Upon depletion of GORAB (Fig. 8a), ST-HRP exhibited a shift in distribution towards the TGN, as well as additional circular profiles within the vicinity of the TGN (Fig. 8b, c). This effect upon ST-HRP distribution was not due to a change in Golgi morphology, which was unaffected by GORAB depletion in these cells (Fig. 8b). Like GORAB, depletion of Scyl1 (Fig. 8a) also resulted in a shift of ST-HRP to later compartments (Fig. 8b, c). As a positive control, we also depleted the Cog3 subunit of the conserved oligomeric Golgi (COG) complex (Fig. 8a), which is a tethering complex required for COPI-dependent Golgi enzyme recycling[45]. As reported previously[49], Cog3 depletion caused extensive vesiculation of Golgi membranes, with a certain proportion of the vesicles containing ST-HRP, indicating a failure to tether *trans*-Golgi-derived vesicles (Fig. 8b, c). In summary, these results reveal that both GORAB and Scyl1 are required to maintain a normal ST-HRP distribution within the Golgi apparatus, as would be expected if they functioned together in COPI-mediated enzyme recycling at the *trans*-Golgi.

**Altered Golgi morphology upon loss of GORAB**. Although the Golgi appeared morphologically normal in GORAB-depleted HeLa cells, we wanted to determine whether the Golgi organization is altered by loss of GORAB in dermal fibroblasts, which represent a better model of the human disease. Dermal fibroblasts

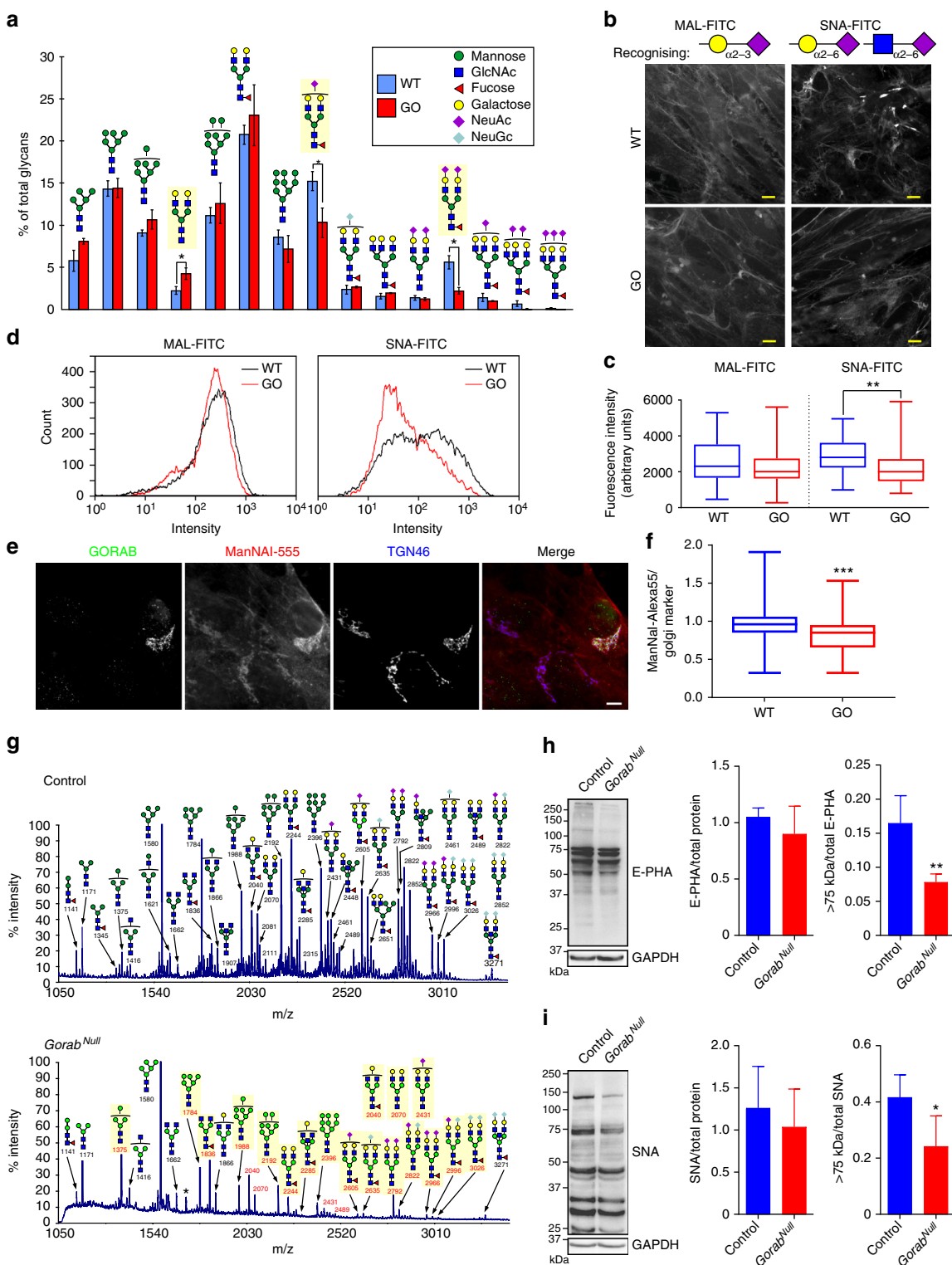

secrete high amounts of extracellular matrix proteins and may therefore be more sensitive to perturbation of intra-Golgi traffic than HeLa cells, which have a lower secretory capacity. As expected, in control fibroblasts, the Golgi apparatus formed a characteristic Golgi stack, with clearly discernable cisternae surrounded by small spherical profiles that likely correspond to transport vesicles (Fig. 8d). Although some cisternal distensions were observed in control fibroblasts, in GO fibroblasts, the distensions were larger and more numerous, and were restricted to one side of the Golgi, most likely the TGN and the *trans*-most

cisternae, where they were often present at the rims (Fig. 8d, e). Hence, GORAB appears to be required to maintain normal organization of the *trans*-Golgi in dermal fibroblasts, which is consistent with a role for the protein in intra-Golgi trafficking at this compartment.

## Discussion

In this study we have identified GORAB as a factor in COPI trafficking at the Golgi apparatus. Together with the COPI

**Fig. 7** Loss of GORAB causes defective terminal *N*-glycosylation of proteins. **a** *N*-glycome analysis of WT and GO fibroblasts. Quantification of relative intensities of MALDI-TOF-MS signals for *N*-glycan species detected in lysates from wild-type (*N* = 3 cell lines) and GO fibroblasts (*N* = 4 cell lines). Error bars represent the mean ± SEM from 4 independent experiments, *p < 0.05, unpaired *t*-test. GlcNAc *N*-acetylglucosamine, NeuAc *N*-acetylneuraminic acid, NeuGc *N*-glycolylneuraminic acid. Yellow shading indicates differences between WT and GO fibroblasts. **b** Analysis of sialylated plasma membrane proteins in WT and GO fibroblasts using MAL and SNA lectins. Top, glycan chains recognized by the lectins. Bottom, non-permeabilized fibroblasts stained with FITC-conjugated lectins. Scale bar, 10 μm. **c** Quantification of fluorescence intensities from **b** (150 cells analyzed per cell line in each of 3 independent experiments, min to max box and whisker plot, **p < 0.01, Mann–Whitney *U* test. **d** Representative flow cytometry histogram of WT and GO fibroblasts (*N* = 3 cell lines) stained with FITC-conjugated MAL and SNA lectins. **e** Analysis of metabolic labeling of WT and GO fibroblasts with alkynyl-tagged sialic acid precursor ManNAl. Co-cultured WT and GO cells were incubated with ManNAl for 10 h, fixed and labeled with antibodies to GORAB and TGN46. Scale bar, 10 μm. **f** Quantification of ManNAl labeling assessed as fluorescence intensity against that of a Golgi marker, with 300 cells analyzed per cell line in 3 independent experiments, min to max box and whisker plot, ***p < 0.001, Mann–Whitney *U* test. **g** *N*-glycome analysis of control and *Gorab*^Null mouse skin tissue. Symbols representing monosaccharide residues are as in **a**. Yellow shading indicates *N*-glycans different between control and *Gorab*^Null samples. **h**, **i** Left, lectin blot analysis of skin lysates of control and *Gorab*^Null E18.5 embryos with E-PHA (**h**) or SNA lectin (**i**). Right, quantification of E-PHA (**h**) or SNA (**i**) levels. Error bars represent the mean + SD, *n* = 4 independent experiments, *p < 0.05, **p < 0.01, unpaired *t*-test. In **a**, **g**, the glycan colored symbols are drawn according to the Symbol Nomenclature For Glycans convention. The structures shown are those most probable for compositions determined from accurate *m/z* measurements on the basis of the well-accepted biosynthetic route for *N*-glycans. Glycan assignments and accompanying masses in **g** are shown in Supplementary Table 1

binding protein Scyl1, it scaffolds COPI assembly at discrete regions (domains) of the *trans*-Golgi. The GORAB domains are functionally important as their loss, or their inability to interact with Scyl1 and therefore COPI, causes GO in humans. The GORAB domains are restricted to the *trans*-Golgi despite COPI being more abundant at the *cis*-side of the Golgi apparatus and the ERGIC. This observation suggests that COPI requires an extra degree of organization to function efficiently at the *trans*-Golgi compared to earlier in the secretory pathway. The TGN is a complex compartment with multiple functional domains[50,51]. Moreover, the predominant Arf binding coat proteins at the TGN are AP1 and the GGAs[51]. Hence, GORAB may improve the efficiency of COPI assembly at the TGN by recruiting Scyl1 and GTP-loaded Arf1, both of which bind COPI, into discrete domains (Fig. 8f). The high local concentration of Scyl1 and Arf1 in these domains would allow the coincident detection of both proteins, and favor the selective concentration of COPI, at the expense of AP1. Because GORAB and Scyl1 are oligomers, recruitment of COPI is likely to be further enhanced by the multivalent nature of the interactions between these proteins. The ability of Scyl1 to bind two distinct sites in COPI also suggests that it may contribute to the coat assembly process by potentially bridging individual coatomer complexes[33].

GORAB is required for recruitment of Scyl1 to the *trans*-Golgi. However, Scyl1 is also present at the *cis*-Golgi and ERGIC[34], and recruitment of Scyl1 to these earlier compartments is independent of GORAB. There are therefore two distinct pools of Scyl1 in the cell, a GORAB-dependent *trans*-Golgi pool and a separate GORAB-independent *cis*-Golgi/ERGIC pool. How Scyl1 is recruited to the *cis*-Golgi and ERGIC is currently unknown. We have shown that it binds to GTP-loaded Arf1, but the persistent association of Scyl1 with the ERGIC upon BFA treatment indicates that another binding factor must exist[34]. A potential candidate is FTCD/58K, which can bind Scyl1[52], but this protein appears to be absent from the ERGIC, suggesting that another, as yet unidentified, protein recruits Scyl1 to this compartment. Regardless of how Scyl1 is recruited to the membrane, its role in promoting COPI assembly appears to be conserved at the different locations. Scyl1 may therefore act as a COPI 'receptor', as has been proposed for p23[15,19]. These proteins may even act in tandem at the *cis*-Golgi and ERGIC to promote COPI recruitment, whereas at the *trans*-Golgi, Scyl1 presumably acts independently of p23, which is not present there.

The other two members of the Scyl family, Scyl2 and Scyl3, are both also present at the Golgi apparatus[43,53,54]. Scyl2, also known as CVAK104, functions as a clathrin adaptor at the *trans*-Golgi, and participates in sorting of SNAREs into clathrin-coated vesicles that shuttle between the TGN and endosomes[53,55]. Its function is therefore distinct from that of Scyl1. In contrast to Scyl2, but similar to Scyl1, Scyl3 appears able to bind COPI, and knockout studies in mice indicate functional redundancy between the Scyl1 and Scyl3, at least in neurons[43]. Hence, these proteins may share overlapping functionality. However, our results, which show that neither Scyl2 nor Scyl3 can bind GORAB, indicate that Scyl1 has a function distinct from that of Scyl3, namely in the scaffolding of COPI into discrete membrane domains at the *trans*-Golgi. It will be interesting to further analyze the role of Scyl3 in COPI traffic, and also to compare how the loss of Scyl1 or Scyl3 affects COPI trafficking in different cell types.

Loss of Scyl1 in humans manifests as CALFAN syndrome, which causes neurodegeneration, similar to that seen in Scyl1-deficient mice[56] and liver failure, with some patients also showing skeletal abnormalities[57,58]. The different symptoms in CALFAN syndrome compared to GO could be explained by Scyl1 functioning earlier in the secretory pathway, at the ERGIC and *cis*-Golgi, in addition to its GORAB-specific function at the *trans*-Golgi[34,52]. This is likely to differentially affect secretory traffic, which may be further complicated by differences in the extent to which loss of Scyl1 or GORAB affects trafficking in different cell types, also considering the possible functional overlap with Scyl3[43]. A better understanding of the disease mechanisms in CALFAN and GO patients will help resolve these issues.

We observed impairment of protein glycosylation in GO cells and in a GORAB knockout mouse. Thus, GO can be considered as a congenital disorder of glycosylation (CDG)[59]. Type II CDGs are associated with defects in glycan processing[59], and we propose that GO is included in this category of CDGs. Interestingly, several type II CDGs are due to mutations in the COG complex, which is required for tethering of intra-Golgi transport vesicles[60]. Loss of COG leads to impaired enzyme recycling, resulting in improper cargo protein glycosylation[45]. Although there is some variability in the severity of the phenotype depending upon the nature of the COG mutation, CDGs due to COG mutations tend to be more severe than GO[45], reflecting the more widespread role for COG in enzyme recycling throughout the Golgi stack. Loss of GORAB tends to cause a milder phenotype, as would be expected from its exclusive role in the recycling of *trans*-Golgi enzymes. Interestingly, wrinkled and lax skin, as seen in GO, is also evident in autosomal recessive cutis laxa type 2 (ARCL2), which is caused by mutation of the ATP6V0A2 subunit of the vacuolar ATPase[61]. The vacuolar ATPase is required to maintain an acidic intraluminal Golgi pH that is optimal for cargo protein glycosylation. Hence, the increased intraluminal Golgi pH upon loss of ATP6V0A2 is thought to cause impaired glycosylation and

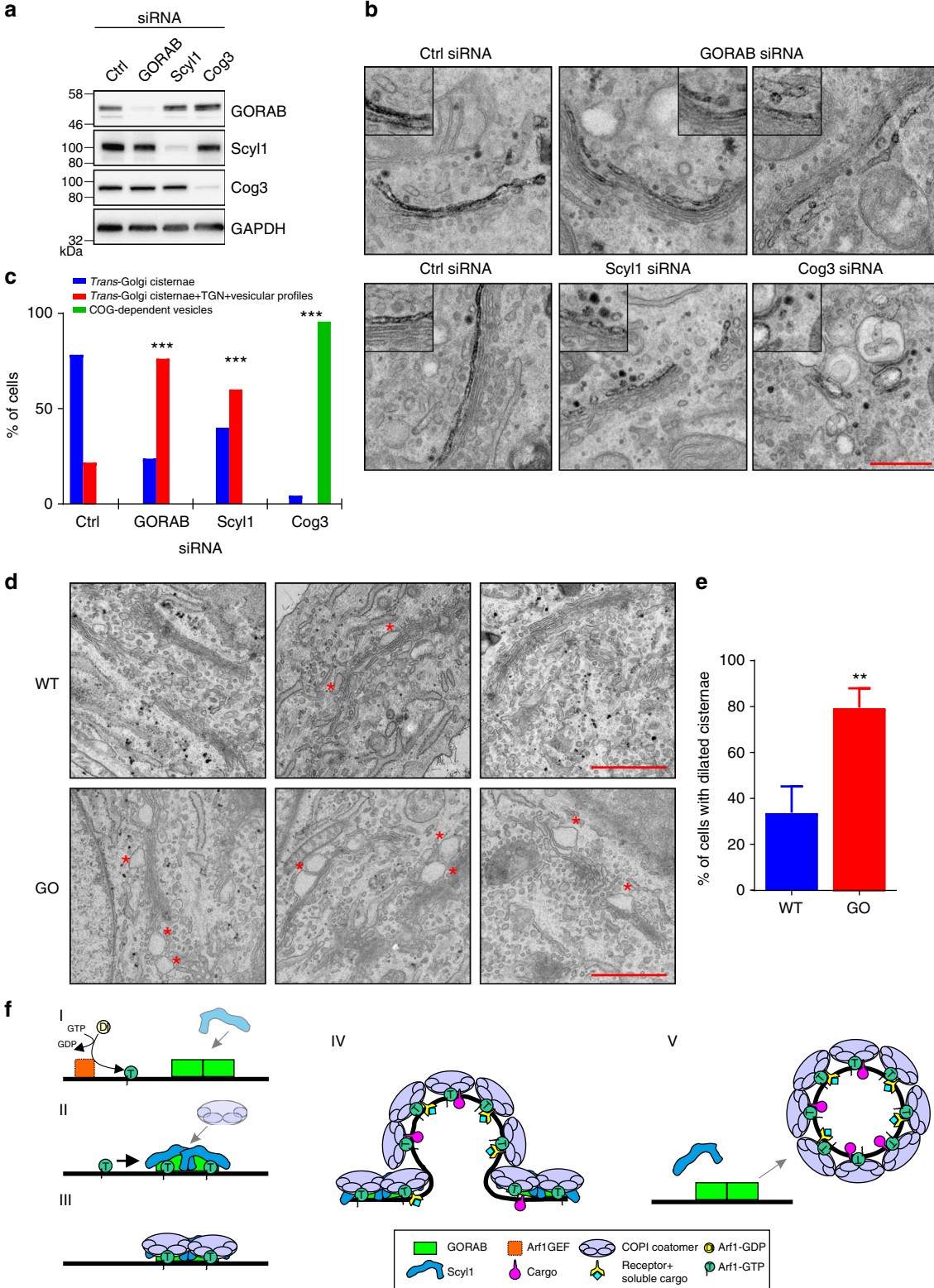

trafficking of secretory cargoes[61,62]. As seen in GO, sialylation of cargo proteins is particularly affected by loss of ATPV0A2[61], suggesting with a common pathogenic mechanism in both ARCL2 and GO.

Although we show here that the modification of N-linked glycans is impaired by loss of GORAB, it is likely that *trans*-Golgi enzymes involved in the modification and processing of O-linked glycan chains, such as those found in proteoglycans, is also affected. Indeed, our analysis of the small leucine-rich

proteoglycans (SLRPs) decorin and biglycan in the skin and bone of GORAB-deficient mice has shown a dramatic reduction in the degree of glycanation of these proteins[47]. SLRPs are abundant proteins of the extracellular matrix, where they associate with collagen to stabilize matrix assembly[63]. Loss of SLRPs causes pathological changes in skin, skeleton and cardiovascular tissues in mouse models and human patients[63]. SLRPs are particularly sensitive to mutation of enzymes involved in glycosaminoglycan (GAG) chain synthesis[64–67], and defects in several of these

**Fig. 8** Loss of GORAB alters SialylT localization and Golgi ultrastructure. **a** Knock-down of GORAB, Scyl1 and Cog3 proteins in HeLa SialylT-HRP cells. SiRNA-transfected HeLa SialylT-HRP cells were lysed, subjected to SDS-PAGE and blotted for GORAB, Scyl1, Cog3 and GAPDH. **b** Representative EM micrographs depict localization of SialylT-HRP, detected using the DAB reaction to generate electron dense product, in HeLa SialylT-HRP cells transfected with the indicated siRNAs. Scale bar, 500 nm. **c** Quantification of SialylT-HRP distribution in siRNA-treated HeLa SialylT-HRP cells ($n = 22$ cells per condition, ***$p < 0.001$, chi-square test). **d** Representative conventional thin section EM micrographs of Golgi ultrastructure in WT ($n = 4$ cell lines) and GO fibroblasts ($n = 3$ cell lines). Enlarged profiles within Golgi cisternae are marked with a red asterisk. Scale bar, 500 nm. **e** Quantification of cells with dilated cisternae. Error bars represent mean ± SD, $n = 25$ cells per cell line, **$p < 0.01$, chi-square test. **f** Proposed model for GORAB function in COPI-mediated trafficking at the *trans*-Golgi. (I) GORAB oligomers are stably associated with the *trans*-Golgi membrane, forming discrete domains, while GTP loading of Arf GTPase leads to its association with the membrane. (II) Membrane-associated GORAB oligomers recruit Scyl1 and locally concentrate GTP-bound Arf in the domains, facilitating the efficient recruitment of coatomer by coincident detection. (III) Coatomer accumulates in the domains and begins to self-assemble. (IV) Coatomer assembly leads to cargo incorporation into a newly forming COPI vesicle. (V) GORAB may stabilize coatomer assembly by remaining associated with the bud neck during vesicle formation. (V) The completed COPI vesicle detaches from the membrane alongside Scyl1, while GORAB stays at the membrane ready to initiate the biogenesis of a new COPI vesicle

enzymes cause connective tissue disorders with similar clinical features to those seen in GO[64,66,67]. It is therefore likely that impaired glycanation of decorin and possibly other proteoglycans, due to defective recycling of glycanation enzymes, contributes to the skin and bone phenotypes seen in GO.

GORAB is widely expressed in the body[22], and we show here that it functions in a universally important process, namely COPI-mediated intra-Golgi trafficking. This raises the question as to why GO is manifest in the skin and bones. One possibility is that loss of GORAB is compensated for by another protein in most tissues. However, GORAB does not have any obvious functional homologs, arguing against this possibility. Rather, we favor the idea that the tissues most affected by loss of GORAB, i.e., skin and bone, are those that are most sensitive to impaired glycosylation and glycanation of cargo proteins. These tissues comprise large amounts of extracellular matrix, and matrix assembly and maintenance are susceptible to impairment of matrix protein glycosylation and glycanation. Hence, loss of GORAB manifests primarily in these matrix-rich tissues. We have shown that decorin is a relevant substrate in this regard[47], but other matrix proteins are also likely to be affected, especially those that undergo extensive sialylation or glycanation.

A recent study using both *Drosophila* embryos and human tissue culture cells has uncovered a role for GORAB in centriole duplication, which is distinct from its function at the Golgi apparatus[31]. This suggests that centriolar defects may contribute to the GO phenotype. However, analysis of a pathogenic GO mutation that disrupts Golgi targeting (A220P) showed no effect upon GORAB function at the centriole. This finding is consistent with Golgi dysfunction being the primary cause of GO, although we cannot exclude an involvement of centriolar defects in GO pathology, possibly through defects at the cilium[30,31]. Interestingly, interference with Golgi targeting of *Drosophila* Gorab resulted in a spermatogenesis defect very similar to that seen in COPI-deficient flies, consistent with a functional association between GORAB and COPI being conserved in evolution[31,68].

## Methods

**Reagents and antibodies**. Reagents were obtained from Sigma-Aldrich, Merck or Thermo Fisher Scientific unless otherwise specified. Primary antibodies used in this study are detailed in Supplementary Table 2. Alexa 488-conjugated streptavidin, Alexa 488-, 546-, 555-, 594- and 647-conjugated, and Cy3- and Cy5-conjugated secondary antibodies were from Molecular Probes (Thermo Fisher Scientific) and from Jackson ImmunoResearch Laboratories, respectively. HRP-conjugated secondary antibodies were from Sigma. HRP-conjugated streptavidin was from GenScript.

**Molecular biology**. GORAB and Scyl1 cDNA sequences were obtained from the I. M.A.G.E. Consortium (Source Biosciences). All amino acid positions of GORAB mentioned in this study refer to the 369 amino acid protein, which originates from the ENST00000367763.7 transcript using the second predicted start codon, which is the correct translation start site[22,25]. Using standard molecular biology techniques full-length and truncated GORAB and Scyl1 sequences were subcloned into pEGFP-

C3 (Clontech Laboratories), pGADT7 and pGBKT7 (BD Biosciences), pFAT2 (a modified pGAT2 vector) and pMAL-C2 (New England Biolabs) for mammalian expression, yeast two-hybrid analysis, and bacterial expression, respectively. Missense patient mutations were introduced by site-directed mutagenesis performed using PfuTurbo DNA polymerase adapted from the Quikchange site-directed mutagenesis method (Agilent Technologies). To make GORAB-mycFKBP constructs, GORAB and mycFKBP fragments were inserted into pcDNA3.1 vector (Invitrogen). Vectors encoding GST-tagged Δ14Arf1 (Q71L and T31N) were a gift from Dr. Sean Munro (Laboratory of Molecular Biology, Cambridge, UK). Arf1 was subcloned into pET24a (Merck) and pcDNA3.1 HA-tag (Invitrogen). GST-tagged Δ14Arf3-GTP (Q71L) and GDP (T31N), Δ14Arf4-GTP (Q71L) and GDP (T31N) and Δ14Arf5-GTP (Q71L) and GDP (T31N) were subcloned from vectors obtained from Dr. Elizabeth Sztul (University of Alabama, Birmingham, USA). Vectors encoding GST-tagged γ−1 appendage, Rab6-GTP (Q72L) and GDP (T27N), Bet1 and syntaxin-1 have been described previously[69–71]. pSRα-SialylT-HRP plasmid containing cytoplasmic tail, transmembrane domain and part of luminal domain of ST6GAL1 fused with HRP has been previously described[48]. Mito-FRB plasmid was a gift from Dr. Stephen Royle (University of Warwick, Warwick, UK). Vector encoding GFP-Scyl2 was obtained from Dr Ernst Ungewickell (Hannover Medical School, Hannover, Germany). Vector encoding Scyl3-myc was obtained from Dr. Rick Thorne (Newcastle, New South Wales, Australia). Scyl3 was subcloned into pEGFP-N3 (Clontech Laboratories). Primer sequences used for molecular cloning are described in Supplementary Table 3.

**Cell culture, transfection, RNAi and drug treatments**. Written informed consent for molecular studies was obtained from control and affected individuals or from their legal representatives. Dermal fibroblasts were obtained by standard punch biopsy. All studies on patient fibroblasts were carried out in accordance with local ethical regulations, with approval from the University of Manchester Research Ethics Committee. Patient fibroblasts were also obtained from the cell line and DNA Bank from Patients Affected by Genetic Diseases (Genova, Italy, codes: FFF0631984 and FFF0731991). All cells were grown at 37 °C and 5% CO$_2$. HeLa (ATCC CCL-2), HeLaM (RRID:CVCL_R965), HEK293 (Cell Biolabs, LTV-100) and human dermal fibroblasts were grown in Dulbecco's modified Eagle's medium (DMEM) supplemented with 10% (vol/vol) fetal bovine serum (FBS), 1 mM L-glutamine and penicillin–streptomycin mix. Non-essential amino acid solution was added to human skin fibroblasts, while HeLa cells stably expressing ST6GALI-HRP (ST-HRP) and HeLa cells stably expressing GFP-GalNAc-T2 (Dr. Brian Storrie, University of Arkansas for Medical Sciences, Little Rock, AK) were supplemented with 1 mg/mL and 0.5 mg/mL G418, respectively. hTERT-RPE-1 cells (ATCC) were grown in 1:1 mix of Ham's F12 and DMEM supplemented with 10% (vol/vol) FBS, 1 mM L-glutamine, penicillin–streptomycin mix and 10 μM hygromycin B. Transient transfection of plasmid DNA was performed using FuGene HD (Promega) according to the manufacturer's instructions and cells were assayed 24–48 h post transfection. For RNA interference (RNAi), HeLa ST-HRP cells were transfected with 20 nM siRNA duplexes using INTERFERin (Polyplus Transfection) according to the manufacturer's instructions and were analyzed 72 h post transfection. GORAB was targeted with ON-TARGETplus SMARTpool (pool of four siRNAs; L-016142; sense: AGCUAGAUAUACAGCGCAA, CAACAACUUCAGC GAGAAA, CAACAAGAACAACGGCUAA and CCAUGAAACUAAAGCGG AU), Scyl1 with ON-TARGETplus SMARTpool (pool of four siRNAs; L-005373, sense: GCUCUGCGGUCUCACUGUA, GAAGUGGUCAGCAGACAUG, CAAG UGAGCCGUGCUAGUC and GCUACACCAGAUCGUGAAA), and Cog3 with a previously described siRNA (sense: AGACUUGUGCAGUUUAACA[49]). all purchased from Dharmacon (Thermo Fisher Scientific). Luciferase siRNA (GL2; Eurogentec, sense: CGUACGCGGAAUACUUCGA) was used as negative control. For the mitochondrial relocation assay, HeLaM cells were treated with 2.5 μg/mL nocodazole for 2 h, followed by addition of 1 μM rapamycin (Calbiochem) for 3 h to induce targeting of GORAB K190del-mycFKBP onto mitochondrial outer membranes. In some experiments cells were incubated with 5 μg/mL brefeldin A (Sigma) for an indicated time period.

**Lentivirus production**. HEK293 cells were seeded on 10 cm dishes 24 h prior to transfection. For each dish, 6 μg of pXLG3-GORAB plasmid, 4.5 μg of psPAX2 packaging plasmid and 3 μg of pM2G envelope plasmid were transfected into HEK293 cells using 27 μL of polyethylenimine mix (1 mg/mL in 150 mM NaCl) and antibiotic-free medium. At 6–8 h after transfection, the medium was replaced. The following day, transfected cells were supplemented with 100 μL of 1 M sodium butyrate (Merck) for 6–8 h and the medium was replaced. At 72 h after the initial transfection, the virus-containing medium was collected and precleared by centrifugation (10 min, room temperature (RT) at $2700 \times g$ in Rotofix 32 A centrifuge (Hettich Centrifuges)) and the supernatant was filtered through a 0.44 μm syringe-driven filter unit. Then, 1–3 mL of virus-containing medium was used for cell transduction.

**Protein-binding assays**. Cells were lysed in HMNT buffer (20 mM HEPES-KOH, pH 7.4, 5 mM MgCl₂, 0.1 M NaCl, 0.5% (wt/vol) Triton X-100) supplemented with protease inhibitor cocktail (Calbiochem) and precleared by centrifugation at $16,000 \times g$ for 15 min at 4 °C in a microfuge. For pull-down experiments, 40 μg of GST-tagged bait protein bound to 20 μL of glutathione resin was incubated with cell lysate (300–500 μL of lysate for exogenously expressed proteins; 2–3 mg of sHeLa cell lysate for endogenously expressed proteins) for 4 h at 4 °C with agitation. Bound proteins were eluted in SDS sample buffer analyzed by sodium dodecyl sulfate–polyacrylamide gel electrophoresis (SDS-PAGE) with western blotting. Proteins from rat liver Golgi membranes[72] were extracted in HKMT buffer (20 mM HEPES-KOH, pH 6.8, 160 mM KOAc, 1 mM MgCl₂, 0.5% (wt/vol) Triton X-100), precleared by centrifugation at $55,000 \times g$ for 10 min at 4 °C and the supernatant was used for pull-down reactions, as described above. For direct binding assays between GST-tagged and MBP-tagged proteins, 20 μg of GST-tagged bait protein bound to 20 μL of glutathione resin was incubated with 20 μg MBP-tagged protein in HMNT buffer supplemented with 100 μg/mL bovine serum albumin for 4 h at 4 °C with agitation. Bound proteins were eluted from the glutathione beads with elution buffer (50 mM Tris-Cl pH 8.1, 25 mM reduced glutathione) for 10 min, followed by trichloroacetic acid precipitation and analyzed by SDS-PAGE with western blotting or Coomassie blue staining. Uncropped versions of western blots are shown in Supplementary Figure 9.

**Surface plasmon resonance**. Experiments were performed using the ProteOn XPR36 instrument (Bio-Rad Laboratories) using the high-capacity GLH chip (Bio-Rad). Running buffer was 150 mM NaCl, 10 mM HEPES, 0.02% (wt/vol) Tween-20, pH 7.4. Two channels were activated with 250 μL of 25 mM N-ethyl-N'-(3-dimethylaminopropyl) carbodiimide (EDC) and 8 mM sulfo-N-hydroxysuccinimide (sulfo-NHS) at a flow rate of 30 μL/min. Anti-MBP antibody was bound to both channels to a final level of approximately 16,000 response units (RUs). MBP-tagged Scyl1 was then captured on the second channel only to a final level of 3000 RUs. Binding of GST-tagged GORAB variants to both channels at 30 nM concentration and a flow rate of 100 μL/min was allowed to occur for 120 s followed by 600 s disassociation, using the first channel as a reference. All binding sensorgrams were collected, processed and analyzed using the integrated ProteOn Manager software (Bio-Rad Laboratories).

**Liposome recruitment assay**. All lipids were purchased from Avanti Polar Lipids. To make 3 mM final 'Golgi lipid' mixture in CHCl₃, the following lipids were used: 43 mol% phosphatidylcholine from bovine liver, 19 mol% phosphatidylethanolamine from bovine liver, 5 mol% phosphatidylserine from bovine brain, 10 mol% phosphatidylinositol from bovine liver, 7 mol% sphingomyelin from bovine brain and 16 mol% cholesterol from wool grease. The liposome suspension was then subjected to five cycles of freezing and thawing using dry ice in isopropanol and 37 °C water bath. For a single experiment, 500 μM liposomes were rehydrated in assay buffer (50 mM HEPES pH 7.2, 120 mM KOAc, 1 mM MgCl₂) and sized via extrusion through a polycarbonate filter with a pore size of 200 nm (GE Healthcare). Liposomes were then incubated at 37 °C for 20 min with 10 μM MBP-Scyl1 or MBP-IPIP27A, 10 μM recombinant mouse coatomer isotype γ2ζ1 (CMγ2ζ1; produced in Sf9 insect cells[18]) and 5 μM recombinant N-myristoylated human ARF1 (purified to near homogeneity), some additionally supplemented with 100 μM GTPγS in a final volume of 100 μL. Next, samples were adjusted to 35% (wt/wt) sucrose, overlaid with 300 μL 30% (wt/wt) sucrose and buffer and centrifuged for 1 h at $256,000 \times g$ in a SW60 rotor (Beckman Coulter). The top fraction (100 μL) containing liposomes was collected, diluted in 500 μL assay buffer, pelleted in a TLA55 rotor (Beckman Coulter) for 1 h at $91,000 \times g$ and analyzed by SDS-PAGE with western blotting.

**Immunofluorescence microscopy**. Cells were grown on glass coverslips and washed twice with phosphate-buffered saline (PBS) prior to fixation in 3% (wt/vol) paraformaldehyde (PFA) in PBS for 20 min at RT. Cells were then washed with PBS and the excess of PFA was quenched with glycine. The cells were permeabilized by 4 min of incubation in 0.1% (wt/vol) Triton X-100 in PBS or in 0.05% (wt/vol) SDS in PBS. Cells were incubated with primary antibody solution for 1 h at RT and incubated three times with PBS for 5 min. Then, coverslips were incubated for 1 h with secondary antibody solution (often supplemented with 200 ng/mL of the DNA dye Hoechst 33342) and incubated three times with PBS for 5 min and

twice in ddH₂0 for 5 min. Coverslips were dried before mounting in Mowiol 4–88 (0.1 M Tris-Cl, pH 8.5, 10% (wt/vol) Mowiol 4–88, 25% (wt/vol) glycerol). Prepared slides were analyzed using an Olympus BX60 upright microscope equipped with a MicroMax cooled, slow-scan CCD camera (Princeton Instruments) driven by Metaview software (University Imaging Corporation). Images were processed using ImageJ software (MacBiophotonics).

**Stimulated emission depletion (STED) microscopy**. Cells were grown on precision glass coverslips (No. 1.5H; Paul Marienfeld), fixed and stained as described above. Images were collected on a Leica TCS SP8 AOBS inverted gSTED microscope using a 100×/1.40 Plan Apo objective. The confocal settings were as follows: pinhole 1 Airy unit, scan speed 400 Hz unidirectional and format 2048 × 2048. STED images were collected using hybrid detectors with the following detection mirror settings: Alexa 488: 498–542 nm; Alexa-549: 564–619 nm; Alexa-647:646–713 nm using the 490 nm, 555 nm and 635 nm excitation laser lines and 592 nm, 660 nm and 775 nm depletion laser lines, respectively. STED images were collected sequentially and deconvolved using Huygens Professional (Scientific Volume Imaging).

**Fluorescence recovery after photobleaching**. HeLa GFP-GalNAc-T2, HeLaM GFP-GORAB or HeLaM cells transiently expressing GFP-Scyl1 were grown in 35 mm glass bottomed dishes (MatTek Corporation). The medium was changed to CO₂-independent medium supplemented with 10% FBS and 1 mM L-glutamine just before FRAP analysis. Images were acquired using a CSU-X1 spinning disc confocal (Yokagowa) on a Zeiss Axio-Observer Z1 microscope with a 150×/1.45 numerical aperture oil immersion TIRF objective (Olympus), Evolve EMCCD camera (Photometrics) and motorized XYZ stage (Applied Scientific Instrumentation). The 488 nm laser was controlled using an AOTF through the laserstack (Intelligent Imaging Innovations) allowing both rapid 'shuttering' of the laser and attenuation of the laser power. FRAP was carried out at 37 °C using the FRAP imaging module of the Slidebook application (Intelligent Imaging Innovations). A 5 μm rectangular region of interest was defined and photobleached at a high laser power to result in >80% reduction in fluorescence intensity. Recovery was monitored by measuring fluorescence intensity at 3 s intervals for a total period of 3 min. FRAP recovery curves were analyzed using FRAPAnalyser software (http://actinsim.uni.lu/; University of Luxembourg, Luxembourg).

**Metabolic labeling with alkyne-tagged sialic acid**. ManNAl was synthesized according to optimized procedures[46]. Cells were grown in DMEM supplemented with 10% FBS and 1 mM L-glutamine containing 500 μM of ManNAl for 10 h before fixation of cells with 4% (wt/vol) PFA. Cells were permeabilized in PBS with 0.1% Triton X-100 for 4 min and incubated with 100 μL/coverslip of a freshly prepared click solution (100 mM K₂HPO₄, 2.5 mM sodium ascorbate, 150 μM CuSO₄, 0.3 mM BTTAA, 10 μM AzidoFluor 545). Copper-catalyzed azide-alkyne [3+2] cycloaddition (CuAAC) was performed for 45 min in the dark at room temperature with gentle shaking. Cells were then stained with antibodies as described above. Images were acquired on a Ti inverted microscope (Nikon) using a 60×/1.40 Plan Apo objective, Proscan II motorized stage (Prior Scientific) and R6 CCD camera (QImaging). A SpectraX LED light engine (Lumencore), quad dichroic (Semrock) and motorized emission filter wheel (Prior Scientific) with single bandpass filters for FITC, TRITC and Cy5 (Semrock) were used to collect image sequences at each position in the tile. Images were acquired and then aligned and stitched using NIS Elements software (Nikon). These stitched images were then exported as a single TIFF image for further processing in Fiji software. The amount of intra-Golgi incorporated alkyne-tagged sialic acid was measured by comparing fluorescence intensity levels with reference to the Golgi marker TGN46. GORAB staining was employed to discriminate between WT and GO fibroblasts.

**Immunofluorescence-based lectin-binding assays**. The following method was adapted from Willet et al.[73]. Human dermal fibroblasts were grown on glass coverslips to 90% confluency. Cells were rinsed twice with pre-chilled PBS and incubated with it for 15 min in order to prevent endocytosis of glycosylated plasma membrane proteins. Next, cells were incubated in FITC-conjugated MAL or SNA lectin solution (20 μg/mL; Vector Laboratories) for 20 min in the cold room. Coverslips were washed three times with pre-chilled PBS and incubated with pre-chilled 4% (wt/vol) PFA solution prepared in PBS for 20 min. Cells were washed three times with PBS and excess PFA was quenched by addition of glycine. Coverslips were washed with ddH₂O, left at RT to dry and mounted using Mowiol 4–88. Samples were imaged on a Ti inverted microscope (Nikon).

**Flow cytometry**. Human skin fibroblasts were grown on 10 cm dishes to 100% confluency. Cells were washed with pre-warmed PBS and detached using pre-warmed Accutase (Sigma). Next, cells were washed twice with PBS, resuspended in pre-chilled PBS and incubated for 15 min on ice followed by incubation in MAL or SNA lectin solution (20 μg/mL; Vector Laboratories) for 30 min at 4 °C. Next, cells were washed three times with PBS, resuspended in 400 μL of ice-cold PBS and analyzed using a Beckman Coulter Cyan ADP flow cytometer with a 488 nm laser. Propidium iodide was added to exclude non-viable cells from the flow cytometry analysis. Data were analyzed using Summit V4.3 software (Beckman Coulter).

**Electron microscopy (EM)**. For morphological analysis, human skin fibroblasts were grown on glass coverslips and flat embedded. Serial thin sections (60 nm) were cut parallel to the coverslip and sections at approximately equal intervals were imaged with Jeol JEM-1400 microscope operated at 80 kV. Images were acquired with Gatan Orius SC 1000B camera. For pre-embedding immuno-EM, cells were fixed with PLP (periodate-lysine-paraformaldehyde) fixative for 2 h, permeabilized with 0.01% saponin, labeled with anti-GORAB rabbit antibody followed by nano-gold-conjugated anti-rabbit IgG $F_{ab}$-fragments (Nanoprobes), post-fixed with 1% glutaraldehyde and quenched with 50 mM glycine. Nano-gold particles were then intensified using the HQ SILVER Enhancement kit (Nanoprobes, Cat. No. 2012) followed by gold toning in subsequent incubations in 2% NaAcetate, 0.05% $HAuCl_4$ and 0.3% $Na_2S_2O_3 \cdot 5H_2O$. The cells then were processed for EM and imaged as described above. Peroxidase cytochemistry was performed on HeLa SialylT-HRP cells that were seeded on Aclar coverslips (Agar Scientific) and transfected with control, GORAB, Scyl1 or COG3 siRNAs. At 72 h after transfection, cells were fixed with 2% (wt/vol) paraformaldehyde and 1.5% (wt/vol) glutaraldehyde solution made in 0.1 M sodium cacodylate buffer, pH 7.4, for 20 min at room temperature (RT). Samples were then washed twice with 0.1 M sodium cacodylate buffer for 3 min and 5 times with 50 mM Tris-buffer, pH 7.6, for 5 min. Samples were incubated in freshly prepared 0.1% (wt/vol) 3,3'-diaminobenzidine (DAB; TAAB Laboratories Equipment) made in 50 mM Tris-buffer, pH 7.6, and supplemented with 0.0002% (vol/vol) $H_2O_2$ for 30 min at RT protected from light. Samples were washed 3 times with 50 mM Tris-buffer, pH 7.6 for 5 min and twice with 0.1 M sodium cacodylate buffer for 5 min. Sections were cut with Reichert Ultracut ultramicrotome and observed with FEI Tecnai 12 Biotwin microscope at 100 kV accelerating voltage. Images were taken with Gatan Orius SC1000 CCD camera.

**Yeast two-hybrid assays**. Yeast two-hybrid assays were performed using Matchmaker Gold system (Clontech Laboratories). First, bait-containing and prey-containing plasmids were co-transformed into the yeast strain Y2HGold with 500 ng of a bait DNA plasmid and 500 ng of a prey DNA plasmid alongside 50 μg of denatured herring sperm DNA acting as a carrier DNA and plated on double drop-out agar plates (SD/-Leu/-Trp). Three single colonies per test condition were inoculated into 4 mL of liquid SD/-Leu/-Trp medium supplemented with glucose and grown for 2 days at 30 °C with 150 rpm agitation. A 10 μL innoculation loop was used to transfer the liquid yeast culture to a square on a double drop-out (SD/-Leu/-Trp) and quadruple drop-out (SD/-Ade/-His/-Leu/-Trp) agar plates. The plates were incubated at 30 °C and growth was monitored for a period of 7 days. SD/-Leu/-Trp agar plate was used as a growth control and the selective growth on the SD/-Ade/-His/-Leu/-Trp agar plate indicated interaction between the bait and the prey.

**Glycan mass spectrometry**. For profiling of control and GO fibroblasts N-glycans were isolated using filter-aided N-glycan separation (FANGS). Fibroblasts were grown until they were confluent, the medium was removed and cells were washed 6 times with PBS. Cells were scraped in 1 mL PBS using a cell scraper and and centrifuged at 16,000×g for 5 min at 4 °C. The cell pellet was dissolved in 10× volume of lysis buffer (4% (wt/vol) SDS, 100 mM Tris, pH 7.6, 100 mM dithiothreitol) and boiled for 5 min at 95 °C followed by centrifugation at 16,000 × g for 5 min at RT. Urea buffer (8 M in 100 mM Tris-Cl pH 8.5) was added to supernatants at a 10:1 volume ratio and samples were passed through ultrafiltration membranes (Amicon Ultra-0.5, Merck) by centrifugation at 15,000 × g for 10 min at RT. Samples retained above filter membranes were subjected to a series of washes combined with centrifugation at 15,000 × g for 10 min at RT: (1) washed twice with 250 μL of urea buffer, (2) incubated with 300 μL of urea buffer supplemented with 40 mM iodoacetamide for 15 min before centrifugation, (3) washed once with 250 μL of 8 M urea and (4) washed four times with 250 μL of 50 mM $NH_4HCO_3$. Filter membranes were subsequently incubated with 8 U of PNGase F in 100 μL of 50 mM $NH_4HCO_3$ for 16 h at 37 °C followed by centrifugation at 15,000 × g for 15 min at RT and washed twice with 250 μL water. Samples above filter membranes containing released N-glycans were transferred to glass tubes and dried in a vaccum centrifuge (Ultraflex Power Technologies). Permethylation of glycans was performed as follows: samples were dissolved in 600 μL of DMSO, supplemented with 25 mg of NaOH and mixed until completely dissolved. Then, iodomethane was added in the following manner: 375 μL followed by incubation for 10 min at RT, 375 μL followed by incubation for 10 min at RT and 750 μL followed by incubation for 20 min at RT. The reaction was quenched by addition of 1.5 mL of 1 g/mL $Na_2S_2O_3$ solution and 1.5 mL of dichloromethane followed by extensive vortexing. Samples were left undisturbed to allow phase separation and the lower, organic, layer was taken to fresh glass tubes and dried under vacuum. Samples were dissolved in 20 μL of methanol. Then, 2 μL of the sample was mixed with 1 μL of 0.5 M sodium nitrate (in 70% methanol) and 2 μL of 20 mg/mL 2,5-dihydroxybenzoic acid (in 70% methanol). Next, 2 μL of this mix was spotted onto a ground steel MALDI target plate (Bruker) and allowed to air dry. Immediately afterwards, 0.2 μL of ethanol was added to the spot and left to air dry for re-crystallization. Glycans were then permethylated and analyzed by mass spectrometry using a Bruker Daltonics ultraflex III TOF/TOF mass spectrometer equipped with a Smartbeam laser used in positive-ion mode over the m/z range 800–5000, with 4000 laser shots in steps of 800, which were summed to give one spectrum per

spot. The Smartbeam™ laser power was set to 50–65%. The Bruker FlexAnalysis software was used to smooth the data (Savitzky–Golay). Following smoothing, all glycan signal intensities assigned a signal-to-noise of >3 by the software were selected, and those belonging to the same species (same isotopic envelope) were summed to generate a total signal intensity for each glycan species. Total signal intensity for each glycan were normalized to the total glycan signal within a spectrum, and normalized intensities averaged between spectra collected for the same cell line.

For glycan profiling of mouse skin samples, glycans were isolated from E18.5 control and homozygous Gorab[Null47]. The mice were bred with local ethical approval from Landesamt für Gesundheitsschutz und Technische Sicherheit (LaGeTSi), Berlin, Germany (approval number G0213/12). The proteins/glycoproteins were then dialyzed against 50 mM ammonium hydrogen carbonate at 4 °C. After lyophilization, glycoproteins were dissolved in 500 μL of 600 mM Tris/HCl, pH 8.2, and denatured by guanidine hydrochloride (6 M final concentration). The sample was reduced using 1 mg of dithiothreitol and incubated at 50 °C for 2 h. After addition of 6 mg of iodoacetamide, the sample was incubated at room temperature for 90 min in the dark. The sample was then dialyzed against 50 mM ammonium hydrogen carbonate at 4 °C and lyophilized. The reduced carboxyamidomethylated proteins were digested with L-1-tosylamide-2-phenylethylchloromethylketone (TPCK) bovine pancreas trypsin (EC 3.4.21.4, Sigma) with an enzyme-to-substrate ratio of 1:50 (by mass), and the mixture was incubated for 24 h at 37 °C in 50 mM ammonium bicarbonate buffer, pH 8.4. The reaction was terminated by boiling for 5 min before lyophilization. PNGase F digestion was carried out in ammonium bicarbonate buffer (50 mM) for 16 h at 37 °C. The reaction was terminated by lyophilization and the products were purified on C18-Sep-Pak to separate the N-glycans from the de-N-glycosylated peptides. After conditioning the C18-Sep-Pak by sequential washing with methanol (5 ml), and 5% acetic acid (2 × 5 ml), the sample was loaded onto the Sep-Pak and the N-glycans were eluted with 2 ml of 5% acetic acid. N-linked glycans were then permethylated using the sodium hydroxide procedure. MALDI-TOF-MS (matrix-assisted laser desorption/ionization–time-of-flight–mass spectrometry) experiments were carried out on Voyager Elite DE-STR Pro instrument (PersSeptive Biosystem, Framingham, MA, USA) equipped with a pulsed nitrogen laser (337 nm) and a gridless delayed extraction ion source. The spectrometer was operated in positive reflectron mode by delayed extraction with an accelerating voltage of 20 kV and a pulse delay time of 200 ns and a grid voltage of 66%. All the spectra shown represent accumulated spectra obtained by 400–500 laser shots. Sample was prepared by mixing a 1 μL aliquot (5–10 pmol) with 1 μL of 10 mg/mL 2,5-dihydroxybenzoic acid (in 50% methanol).

The assignment of glycan species for both human fibroblast and mouse skin samples was based on accurate m/z measurements, precisely matching to theoretical masses of the glycan species measured, taking into account the known ionization of these glycans, and on the basis of the well-accepted biosynthetic route for N-glycans[74,75].

**Statistical analysis**. Statistical analyses were conducted using GraphPad Prism software (GraphPad Software). D'Agostino–Pearson and Shapiro–Wilk tests were used for comparison of the distribution of data with a Gaussian distribution. Depending on the result, an unpaired t-test or Mann–Whitney test was performed. In the case of an unpaired t-test, equality of variances between two groups was tested with an F-test. One-way analysis of variance with Dunnett's test was performed for multiple group comparisons and the equality of group variances were examined with the Brown–Forsythe test. Quantification of SialylT-HRP distribution in siRNA-treated HeLa SialylT-HRP cells was performed using the chi-square test. Statistical significance cut-offs were set as follows: *$p \leq 0.05$, **$p < 0.01$ and ***$p < 0.001$.

## Data availability
The data that support the findings of this study are available from the corresponding author upon request.

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

## Acknowledgements

T.M.W. was supported by a PhD studentship from the Wellcome Trust (096601/Z/11/Z) and M.L. by an MRC research grant (MR/N000366/1). We thank the Wellcome Trust for equipment grant support to the EM Facility. The Bioimaging Facility microscopes used in this study were purchased with grants from BBSRC, Wellcome and the University of Manchester Strategic Fund. Special thanks go to Peter March for his help with the light microscopy and Mike Jackson from the Flow Cytometry Core Facility for his help and advice. The York Centre of Excellence in Mass Spectrometry was created thanks to a major capital investment through Science City York, supported by Yorkshire Forward with funds from the Northern Way Initiative. U.K. received funding from the EU (E-Rare project EURO-CDG-2) and from the German Federal Ministry of Education and Research (BMBF) (DIMEOs (1EC1402B)). We give thanks to various colleagues for generously providing reagents as indicated in the Methods section, and thank Professor Philip Woodman (University of Manchester) for comments on the manuscript.

## Author contributions

M.L. managed the project and together with T.M.W. conceptualized the experiments. T.M.W. performed all the experiments apart those described below. T.M.W. and M.L. analyzed the data. T.M.W. prepared the figures for the manuscript. M.L. wrote the manuscript with input from all co-authors. M.T. helped conceive the project and provided patient fibroblasts. W.L.C. and U.K. generated the GORAB knockout mouse and prepared tissues from it. M.J. and E.J. generated and analyzed the immuno-EM and fibroblast EM data. A.A.M. performed the EM analysis of the HRP-ST cells. E.P., J.T.-O. and D.U. generated and analyzed the glycomics data for fibroblast cells. W.M. performed the glycan analysis of the GORAB knockout mouse samples. A.P.M. helped perform the surface plasmon resonance experiments. C.B. and Y.G. made and provided expertise in the use of alkyne-tagged sialic acid. M.R. and F.T.W. provided assistance with the liposome experiments.

## Additional information

**Competing interests:** The authors declare no competing interests.

