## [Peer Review File · Nature Communications]

Reviewers' comments:

Reviewer #1 (Remarks to the Author):

This is a high quality study that presents a comprehensive investigation of the function of GORAB, a Golgi protein that is mutated in gerodermia osteodysplastica. The authors use an impressive range of methods including imaging, biochemistry, mass-spectrometry and immuno-EM to arrive at a model in which GORAB forms stable "domains" on the trans-Golgi that recruit a second protein Scyl1, and that together these proteins aid the recruitment of COPI coats. Removing GORAB or Scyl1 causes defects in retention of a trans-Golgi sialyltransferase, and alterations in protein glycosylation. This is consistent with both the known role of COPI in recycling Golgi enzymes in the Golgi stack, and the defects in glycosylation seen in patients with gerodermia osteodysplastica.

The data are generally of a high quality and very well presented in the Figures. The text of the paper is clearly written and easy to read, and the Discussion is thoughtful and measured. Overall I was impressed by this study. However there are a few areas where extra controls or relatively small further experiments or would add a lot to the robustness of the data, and a couple of areas where further discussion is required of other papers in the field. These points are listed below. If they can be addressed I would be very happy to recommend publication of what should be, once revised, a really excellent paper.

Experimental Issues:

- a) Figure 2A. The authors state that GFP-GORAB is punctate, but much of the staining looks continuous.
- b) Figure 2G. The BioID experiment has little value as presented as there is no control of another Golgi protein, nor do they provide a full list of hits to assess the significance of the ones they show. Both of these should be provided, or the data removed.
- c) The authors show that GORAB persists when Scyl1 is depleted. What happens to Scyl1 when GORAB is depleted?
- d) The rapid exchange of Scyl1-GFP upon photobleaching seems a bit hard to reconcile with a stable interaction with forming COPI-coated vesicles. The imaging data should be shown and not just the quantitation, and this problem discussed.
- e) Have the authors examined the location of Rab6 relative to the GORAB "domains"?
- f) The authors examine the effect on GFP-Scyl1 of directing GORAB to mitochondria. What happens to endogenous Scyl1? Does the relocation of endogenous COPI require the co-overexpression of GFP-Scyl1?
- g) In the mitochondrial GORAB experiments it would be good to look at a couple of other Golgi markers to prove that the relocation of COP1 does simply reflect the relocation of much of the Golgi.
- h) Why are only high molecular weight species affected in patient cells (Figure 6F and 6G (note this is incorrectly stated to be 6F and 6I in the text)).
- i) The authors provide some very impressive EM images of ST-HRP in wildtype and siRNA treated cells (Fig 7). They should look to see if more ST-HRP is secreted into the medium in the siRNA treated cells. Many glycosyltransferases are proteolysed and released into the medium and this may increase in the absence of GORAB or Scyl1.

2) Textural issues.

a) The only substantial textural issue is that the authors really must discuss the fact that patients lacking Scyl1 have different symptoms to those lacking GORAB. This may be explained by Scyl1 also having a role elsewhere in the Golgi, but it needs to be discussed. In addition, the authors need to discuss the fact that Scyl1 has a relative called Scyl3 that is also on the Golgi and binds to COPI. There is evidence that these proteins are partially redundant (see PubMed ID 29437892). Is Scyl3 expressed in the cells used in this study? Finally, the authors should briefly discuss Scyl2 (aka CVAK104) which has been proposed to have a role in forming AP-1/clathrin coated vesicles.

b) GORAB and Scyl1 are not "novel" Arf effectors, as this interaction has already been reported.

c) Could GORAB "domains" simply be COPI-coated regions?

d) The first use of rapamycin-induced dimerisation to induce mitochondrial targeting that I am aware of was not Robinson et al 2010, but Silvius et al 2006 (PubMed ID 16236799).

Reviewer #2 (Remarks to the Author):

In the manuscript by Witkos et al the authors examine the functions of GORAB as a scaffold protein that mediates the proper organization and recycling of enzymes in the Golgi complex. GORAB is mutated in geroderma osteodysplastica, a disease associated with loose skin and osteoporosis. Understanding its function could provide insight into this disease. It was known that GORAB localizes to the trans-Golgi and interacts with Arf5 and Rab6 and also potentially interacts with Scyl1. Here they show that GORAB directly interacts with Scyl1 (and map that interaction to their amino terminal domains), that GORAB, Scyl1 and COPI can be found on discrete puncta in the trans Golgi, and that all three are Arf1 effectors. GORAB seems to be more stably associated to the Golgi and its overexpression protects Scyl1 and COPI from BFA-induced dissociation. Disease mutations in GORAB inhibit its binding to Scyl1 and indirectly COPI. Finally, depletion of GORAB leads to altered Golgi morphology, shift of sialyl transferase to the TGN and a reduction in terminal sialylation of glycoproteins. In fibroblasts from patients there are dilated Golgi cisternae. Taken together the data suggest that GORAB serves an important scaffolding function at the trans Golgi to recruit Scyl1 and COPI, presumably for proper recycling of Golgi enzymes. The study is thorough and convincing. There are only a few questions that the authors could address that would add to the strength of their study.

1. Is it known where Rab6 and Arf1 (5) bind to GORAB? Since Scyl1, GORAB and COPI all bind to Arf1 and Scyl1 binds GORAB and COPI - any evidence or data about whether these interactions can occur simultaneously?

2. In Fig. 5 the knock-sideways to mitochondria assay is striking. Was Arf1 also recruited to the mitochondria? I ask since the GORAB K190del mutant used binds to Arf1 especially strongly. This could be done with Arf1-GFP.

3. Fig. 7 could be better annotated and described. How are cis and trans sides of the Golgi determined? I assume that in D antibody to GORAB is shown and it is reduced in the GO samples - fewer black dots(?). Please clarify.

Minor correction: For Figure 4A. Change "Localization of known missense...." to "Location of known missense....."

Reviewer #3 (Remarks to the Author):

In the manuscript, the authors studied GORAB using an array of techniques, including mass spectrometry, lectin-binding assay, flow cytometry and microscopy. It was already reported that GORAB interacts with Scyl1 (Di et al., 2003), and Scyl1 is known to regulate COPI-mediated retrograde protein traffic at the interface between the Golgi apparatus and the endoplasmic reticulum (Burman et al., 2008) and regulates Golgi morphology (Burman et al., 2010; Schmidt W.M, et al. Am. J. Hum. Genet. 97:855-861(2015)). They also "re-evaluated GORAB interaction with Arfs." The class I and II Arf GTPases both promoted membrane recruitment of COPI. It is well known that COPI plays critically important role in vesicle trafficking, and thus it is not surprising that protein glycosylation and glycan structures altered after COPI was perturbed. The author performed thorough studies of the protein GORAB. The results are interesting, but not unexpected. Therefore this manuscript could be submitted to a more specialized journal.

For Fig. 1C and E, proteins were stained with Coomassie Blue. Overall, the signal is too weak. The first lane was loaded with the whole cell lysate, but barely any bands appeared. Even though Scyl1 was overexpressed, supposedly there were many highly abundant proteins in the lysate. On Page 6: "The binding between GORAB and Scyl1 is direct, as indicated by pull-down experiments with purified recombinant proteins (Fig 1C)." It is a bit questionable to conclude that the binding between GORAB and Scyl1 is direct based on the pulled-down experimental results.

On Page 9: "This result indicates that GORAB is stably associated with the domains, and therefore that the domains themselves are stable entities." It seems that the "domain" here is different from the commonly used word "domain", such as the one in Figure 1F (kinase-like domain). Please clarify.

It might be better to describe in more details regarding the glycan assignments in Figure 6.

We would like to thank the reviewers for their constructive comments on our manuscript. Please find our responses to the comments in a point-by-point manner below. Note that the new text in the revised manuscript is highlighted in red.

Reviewer 1:

Experimental Issues:

a) Figure 2A. The authors state that GFP-GORAB is punctate, but much of the staining looks continuous.

We have replaced the previous image of GFP-GORAB with a better quality example that shows more clearly the punctate distribution of the protein at the Golgi. This is present in a new Figure 2B.

b) Figure 2G. The BioID experiment has little value as presented as there is no control of another Golgi protein, nor do they provide a full list of hits to assess the significance of the ones they show. Both of these should be provided, or the data removed.

In light of these valid criticisms we have decided to remove the Bio-ID data from the manuscript. Figures 1D, 2G and Supplementary Figure 1 and accompanying text have therefore been removed from the paper.

c) The authors show that GORAB persists when Scyl1 is depleted. What happens to Scyl1 when GORAB is depleted?

To address this point we have analyzed the distribution of Scyl1 in patient fibroblasts lacking GORAB, comparing it to the distribution of Scyl1 in control cells that contain GORAB. The results show a marked reduction in perinuclear staining and reduced colocalization with p230, in the GORAB-deficient cells, indicating that a significant amount of Golgi-associated Scyl1 requires GORAB for its localization. The residual Scyl1 staining in the GO cells is likely the pool of the protein present at the ERGIC/*cis*-Golgi, whose recruitment is independent of GORAB. The new data is shown in Supplementary Figure 4, and described on page 12 of the main text.

d) The rapid exchange of Scyl1-GFP upon photobleaching seems a bit hard to reconcile with a stable interaction with forming COPI-coated vesicles. The imaging data should be shown and not just the quantitation, and this problem discussed.

This is a good point. Our interpretation is that GORAB forms stable domains onto which Scyl1 dynamically exchanges. This would suggest that the requirement for Scyl1 in COPI recruitment is transient. Interestingly, a similar phenomenon is observed for Arf1, which has a shorter residency time on Golgi membranes compared to COPI (Presley et al, Nature, 2002). Presumably additional interactions of COPI, such as binding to cargo, allow for a more stable association of the coat with the membrane following its initial recruitment. Such a model would fit with a more dynamic membrane exchange of both Scyl1 and Arf1 compared to COPI. To explore the dynamic relationship of GORAB and Scyl1 in more detail, we analyzed the rate of membrane exchange of Scyl1 upon co-expression of GORAB. Under this condition the membrane exchange of Scyl1 was slowed, and the immobile pool also increased, indicating that GORAB levels can influence the rate of exchange of Scyl1 with the membrane. However, Scyl1 exchange remained significantly faster than that of GORAB, consistent with it dynamically exchanging with GORAB domains. This new data is added in Supplementary Fig 2 and described on page 10 of the text.

e) Have the authors examined the location of Rab6 relative to the GORAB “domains”?

We have performed colocalization analysis of Rab6 and GORAB. The results show that Rab6 is more widely distributed throughout the *trans*-Golgi compared to GORAB, consistent with it participating in several functions at this compartment. There is however some colocalization of Rab6 with the GORAB domains, consistent with the previously reported interaction between the proteins. The data is included in a new Figure 2G and described on page 8 of the text.

f) The authors examine the effect on GFP-Scyl1 of directing GORAB to mitochondria. What happens to endogenous Scyl1? Does the relocation of endogenous COPI require the co-overexpression of GFP-Scyl1?

Endogenous Scyl1 is also recruited by GORAB when it is directed to mitochondria. Under these conditions there is little recruitment of endogenous COPI, especially compared to when GFP-Scyl1 is co-expressed. We think this is because endogenous Scyl1 is present in limiting amounts compared to those of COPI, which is more abundant in cells. The new data is shown in Figure 5C and Supplementary Figure 6A and described on page 13 of the text.

g) In the mitochondrial GORAB experiments it would be good to look at a couple of other Golgi markers to prove that the relocation of COPI does simply reflect the relocation of much of the Golgi.

To address this important point we have analyzed the localization of several Golgi markers. None of these are redistributed by mitochondrially-targeted GORAB excluding the possibility of relocation of Golgi elements. The new data is shown in Supplementary Figure 6A and described on page 13 of the text.

h) Why are only high molecular weight species affected in patient cells (Figure 6F and 6G (note this is incorrectly stated to be 6F and 6I in the text)).

We thank the reviewer for this important question. We assume that the sialylation defect induced by loss of GORAB at the Golgi compartment globally affects more or less all proteins carrying complex N-glycans. Higher molecular weight species may be more sensitive if they have more, or more complex, glycan chains with higher numbers of sialic acid residues. In addition, reduced terminal glycosylation could influence migration upon SDS-PAGE. In this case, for the many heavily glycosylated proteins running at apparent molecular weights ≥ 75 kDa, a global reduction of sialylation would result in a lower intensity in the upper part of the lectin blots, while in the lower parts the reduced lectin binding would be compensated by the downward shift of these glycoproteins. A similar effect is seen in proteoglycans in Gorab-deficient tissues, where a global reduction in dermatan sulfate levels leads to a reduced apparent weight of the proteoglycan decorin (Chan et al, PloS Genetics, 2018). The figure number has been corrected in the text.

i) The authors provide some very impressive EM images of ST-HRP in wildtype and siRNA treated cells (Fig 7). They should look to see if more ST-HRP is secreted into the medium in the siRNA treated cells. Many glycosyltransferases are proteolysed and released into the medium and this may increase in the absence of GORAB or Scyl1.

To address this point we analyzed the secretion of the ST-HRP in control versus GORAB, Scyl1, or COG3-depleted cells. In all cases we failed to observe ST-HRP in the medium. This could be explained by the ST-HRP chimera, which contains only a short region of the luminal

stalk domain of ST6GAL1 (amino acids 26-45), lacking the proteolytic site necessary for cleavage and subsequent release from cells, which was identified in an early study as residue 64 (Weinstein et al, JBC, 1987). Although BACE1 has been reported to cleave ST6GAL1 at amino acid 37 (of the rat protein, equivalent to amino acid 40 of the human version) (Kitazume et al, JBC, 2003), our failure to detect the ST-HRP chimera in the medium would suggest it is not sensitive to cleavage by BACE1 in our experimental system. To circumvent any potential issues with cleavage of the chimera, we analyzed endogenous ST6GAL1, for which there are antibodies that work for Western blotting, although unfortunately they do not work for microscopy. In control cells ST6GAL1 was detected in the medium, running as a faster migrating form, as expected. Unexpectedly, we detected less ST6GAL1 in the medium of GORAB or Scyl1-depleted cells. Interestingly, the same effect was seen upon depletion of COG3. Although these results were unexpected, the similarity in phenotype between depletion of GORAB (and Scyl1) and COG3 are consistent with reduced secretion being caused by defective recycling and retention within the Golgi. Perhaps ST6GAL1 is misrouted to the lysosome under these conditions, as opposed to being secreted. Misrouting of *cis*-Golgi enzymes to the lysosome upon inefficient retention in the Golgi has been reported previously (van Meel et al, PNAS, 2014), so there is some precedent for this. Another explanation could also be that the protease(s) responsible for ST6GAL1 cleavage, which also reside in the *trans*-Golgi, are less efficiently retained upon depletion of the GORAB (and Scyl1) and COG3, resulting in less cleavage and less secretion as a consequence. Because we cannot explain the secretion data, we decided not included them in the manuscript, but they are provided below for the referee. We would be happy to include them in the manuscript should the referee think it necessary.

Analysis of sialyltransferase secretion. **A.** Knock-down of GORAB, Scyl1 and Cog3 proteins in HeLa SialylT-HRP cells. SiRNA transfected HeLa SialylT-HRP cells were lysed, subjected to SDS-PAGE and blotted for GORAB, Scyl1, Cog3 and GAPDH. **B.** Analysis of SialylT-HRP secretion in HeLa SialylT-HRP cells. HeLa SialylT-HRP cells were transfected with siRNA for 60 h and grown for additional 12 h in fresh serum-free medium to assess the release of SialylT-HRP to the medium. Cell lysates and TCA-precipitated medium fractions were subjected to SDS-PAGE and blotted for HRP and GM130 as a loading control for cell fractions. **C.** Analysis of endogenous ST6GAL1 secretion in HeLa SialylT-HRP cells. Cells were treated with siRNA and cell and medium fractions were collected as described in B. Samples were subjected to SDS-PAGE and blotted for ST6GAL1 and GM130. **D.** Quantification of normalized ST6GAL1 medium/cells ratio in siRNA-treated SialylT-HRP cells (n=4, * p<0.05, ** p<0.01, *** p<0.001).

Textural issues.

a) The only substantial textural issue is that the authors really must discuss the fact that patients lacking Scyl1 have different symptoms to those lacking GORAB. This may be explained by Scyl1 also having a role elsewhere in the Golgi, but it needs to be discussed. In addition, the authors need to discuss the fact that Scyl1 has a relative called Scyl3 that is also on the Golgi and binds to COPI. There is evidence that these proteins are partially redundant (see PubMed ID 29437892). Is Scyl3 expressed in the cells used in this study? Finally, the authors should briefly discuss Scyl2 (aka CVAK104) which has been proposed to have a role in forming AP-1/clathrin coated vesicles.

The difference in phenotypes between GORAB and Scyl1-deficient patients is an interesting point. We do not know the reason for it, but as the referee indicates, the most likely explanation is Scyl1 operating at different locations within the secretory pathway compared to GORAB. It clearly exists as a GORAB-independent pool at the ERGIC/*cis*-Golgi, so defective trafficking there in addition to the *trans*-Golgi could account for the different phenotype seen in Scyl1-deficient patients compared to GO patients. This point is now discussed within the Discussion section, on pages 22 and 23 of the text. Regarding Scyl3, we cannot exclude the possibility of some functional redundancy with Scyl1. Scyl3 is expressed in our cells. However, we failed to detect binding of Scyl3 to GORAB, using both the mitochondrial relocation assay and protein pull-down experiments, and thus believe Scyl3 is not relevant for the functional associations we describe in the current manuscript. Similarly, we failed to detect binding of Scyl2 to GORAB, indicating that of the Scyl family, it is only Scyl1 that associates with GORAB. The new Scyl2 and Scyl3 binding data is presented in Supplementary Figure 8B and C and described on page 14 of the text. The recent Scyl3 paper describing functional overlap with Scyl1 is mentioned within the text on page 14, and also discussed further on pages 21 and 22, and we also discuss Scyl2 at the Golgi on pages 21 and 22.

b) GORAB and Scyl1 are not “novel” Arf effectors, as this interaction has already been reported.

This point is well taken. With regard to Scyl1 the previous study by Hamlin et al (JCS, 2015) described binding to Arfs in a non-nucleotide-dependent manner, which is contrary to our finding that binding occurs only to the GTP-bound state. The previous study on GORAB by Egerer et al (JID, 2015) did report nucleotide-dependent binding, as expected for an effector protein. We have therefore corrected the relevant text on page 9 to remove the "novel" term.

c) Could GORAB “domains” simply be COPI-coated regions?

We don't think this the case as we can observe GORAB domains that lack COPI staining, and also others that lack staining for both Scyl1 and COPI (Figure 2F). These observations, combined with our other results, lead us to believe that the GORAB domains are stable entities that are able to recruit COPI, via Scyl1, to nucleate COPI assembly.

d) The first use of rapamycin-induced dimerisation to induce mitochondrial targeting that I am aware of was not Robinson et al 2010, but Silviu et al 2006 (PubMed ID 16236799).

We thank the referee for pointing out this mistake. The reference has been corrected.

Reviewer 2:

1. Is it known where Rab6 and Arf1 (5) bind to GORAB? Since Scyl1, GORAB and COPI all bind to Arf1 and Scyl1 binds GORAB and COPI - any evidence or data about whether these interactions can occur simultaneously?

The Rab6 and Arf binding sites in GORAB have previously been mapped to a central coiled-coil region of the protein (Egerer et al, JID, 2015). GO patient mutations differentially affect binding to Rab6 and Arf, indicating that they have distinct binding sites within this region (Egerer et al, JID, 2015), consistent with GORAB being able to bind both proteins simultaneously. To assess whether Scyl1, GORAB, COPI and Arf1 can bind simultaneously, we performed a pull-down experiment including all of these protein components in purified form. The result showed that all proteins could be detected in the bound fraction, with no detectable competition between one protein and another, supporting the view that all of the proteins can interact together to form a complex. This new data is included in Supplementary Figure 1H and described on page 9 of the text.

2. In Fig. 5 the knock-sideways to mitochondria assay is striking. Was Arf1 also recruited to the mitochondria? I ask since the GORAB K190del mutant used binds to Arf1 especially strongly. This could be done with Arf1-GFP.

To address this interesting question, we performed the mitochondria relocation assay using HA-tagged Arf1. Under conditions where we see efficient relocation of COPI, there is very little if any recruitment of Arf1 to GORAB and Scyl1 at mitochondria. This may be due to the fact that in order to be recruited to membranes and engage with its effectors, Arf1 must exchange bound nucleotide via a guanine nucleotide exchange factor, which is presumably lacking from the mitochondrial membrane. The absence of Arf1 at mitochondria further supports the view that GORAB and Scyl1 can promote membrane recruitment of COPI even in the absence of membrane-associated Arf1. The new data is shown in Supplementary Figure 6C and described on page 14 of the text.

3. Fig. 7 could be better annotated and described. How are cis and trans sides of the Golgi determined? I assume that in D antibody to GORAB is shown and it is reduced in the GO samples - fewer black dots(?). Please clarify.

We apologize for the confusion here. In panels B (quantified in C), the *trans*-side of the Golgi is marked by the ST-HRP chimera, which is known to localize to the *trans*-Golgi (Stinchcombe et al, JCB, 1995; Jokitalo et al, JCB, 2001). For panel D, there was no immunolabeling. The small black dots the referee alludes to may correspond to endogenous electron-dense glycogen granules that are present in these cells. The legend has been modified to state more explicitly that the images in panel D are from conventional TEM. In the text we avoid being definitive and use the terms "likely" and "appears to be" to indicate our thinking that the distensions are on this side of the Golgi, which is based on the fact that GORAB is a *trans*-Golgi protein, and thus most likely to affect *trans*-Golgi morphology.

Minor correction: For Figure 4A. Change "Localization of known missense...." to "Location of known missense....."

We thank the referee for spotting this mistake. It has been corrected.

Reviewer 3:

It is well known that COPI plays critically important role in vesicle trafficking, and thus it is not surprising that protein glycosylation and glycan structures altered after COPI

was perturbed. The author performed thorough studies of the protein GORAB. The results are interesting, but not unexpected. Therefore this manuscript could be submitted to a more specialized journal.

In this study we uncover a novel mechanism for scaffolding of the COPI vesicle coat, a key player in protein trafficking within the secretory pathway. The identification of this scaffold, constituted by stable GORAB membrane domains that form a platform for COPI recruitment, is completely new and was not previously foreseen. Thus, although we have known for many years that Arf is the primary driver for COPI recruitment to the Golgi, we now identify a key additional factor that is required for this process *in vivo*, at least at the *trans*-Golgi, and uncover its mechanism of action.

Although it has been known for several years that loss of GORAB function causes disease in humans, a lack of knowledge regarding the protein's function has precluded the identification of a disease mechanism. Our work rectifies this issue by identifying a major cellular role for the protein in COPI vesicle trafficking. Interestingly, a function for GORAB at the centriole has also recently been described (Kovacs et al, Nature Genetics 2018), but importantly, we show in our study that pathogenic patient mutations interfere with GORAB function at the Golgi, strongly supporting the view that Golgi impairment underlies the human disease.

Other novel aspects of our study include the following: a) Although Scyl1 has been shown to bind COPI previously, its function at the Golgi has remained elusive. We now show that it is an important component of the *trans*-Golgi COPI scaffolding machinery, and map out its function in this process. b) Previous work on Scyl1 has localized it exclusively to the *cis*-Golgi and ERGIC, and has also erroneously described the interaction between Scyl1 and Arf as being nucleotide independent. Our work shows that Scyl1 exists in later Golgi compartments, and that Scyl1, like GORAB, is an Arf effector. Both are important when considering Scyl1 cellular function. c) Our work provides strong evidence that COPI functions at the *trans*-Golgi, which has been a matter of some contention in the field (see for example Papanikou and Glick, Curr Op Cell Biol, 2014, and Bykov et al, eLife, 20017). d) Finally, GORAB was originally described as a member of the golgin family of coiled-coil proteins (Hennies et al, Nature Genetics, 2008), suggesting a function in vesicle tethering (golgins function as vesicle tethering proteins at the Golgi). Our work shows that GORAB is not in fact a golgin, and instead functions in COPI recruitment for vesicle formation at the Golgi, as opposed a role in vesicle tethering.

For Fig. 1C and E, proteins were stained with Coomassie Blue. Overall, the signal is too weak. The first lane was loaded with the whole cell lysate, but barely any bands appeared. Even though Scyl1 was overexpressed, supposedly there were many highly abundant proteins in the lysate. On Page 6: “The binding between GORAB and Scyl1 is direct, as indicated by pull-down experiments with purified recombinant proteins (Fig 1C).” It is a bit questionable to conclude that the binding between GORAB and Scyl1 is direct based on the pulled-down experimental results.

We believe there is a misunderstanding here. Only purified recombinant proteins were added to the assay. The first lanes in panels C and D show the purified prey protein (MBP-Scyl1). There are some lower MW bands present at low abundance, but these correspond to degradation products or low amounts of bacterial contaminants. This is now mentioned in the accompanying figure legend, which we have modified. To further show the purity of the proteins used in the experiment, we have included a figure below for the reviewer, showing a Coomassie Blue staining of the fractions from the purification of the proteins, and the final

purified proteins themselves. Because we only added purified proteins to the assay, alongside BSA as a carrier protein, we are confident in our assertion that GORAB and Scyl1 interact directly.

Purification of recombinant MBP-Scyl1 and GST-GORAB after bacterial expression. BL21 codon plus *E.coli* strain was transformed with pMAL-C2 MBP-Scyl1 or pFAT2 GST-GORAB vectors and the liquid cultures (*uninduced*) were induced with 100 μ M IPTG at $OD_{600}=0.6$ and grown overnight at 18°C (*induced*). Bacteria were pelleted and lysed in the lysis buffer (20 mM HEPES-K pH 7.4, 200 mM NaCl, 1 mM DTT, protease inhibitor cocktail, 50 μ g/mL lysozyme) for 30 min on ice (*total lysate*), followed by sonication. Unbroken cells and cell debris were pelleted. *Supernatant* was added to 1 mL of pre-equilibrated glutathione sepharose resin. 10% (weight/vol) Triton X-100 was added to a final concentration of 1% and beads were incubated overnight at 4°C. The bacterial pellet was resuspended in lysis buffer (*pellet*). Beads were washed four times with lysis buffer and applied to an Econo-Pac 10 mL column (Bio-Rad Laboratories, Hercules, CA) column (*wash*) and eluted with lysis buffer supplemented with 50 mM glutathione and collected in 0.5 mL fractions. Peak fractions were pooled and the protein was desalted into PBS using PD-10 desalting columns (GE Healthcare, Little Chalfont, UK) following the manufacturer's instructions (*eluted and desalted*). Unbound fraction (*unbound*) was flash frozen and stored at -80°C. Collected fractions during bacterial purification were mixed with 2x SDS sample buffer, subjected to SDS-PAGE and stained with Coomassie blue to assess the purity of obtained product. Full-length MBP-Scyl1 and GST-GORAB are marked with arrows.

On Page 9: “This result indicates that GORAB is stably associated with the domains, and therefore that the domains themselves are stable entities.” It seems that the “domain” here is different from the commonly used word “domain”, such as the one in Figure 1F (kinase-like domain). Please clarify.

The term "domain" is commonly to describe regions within membranes, as well as regions within proteins. We therefore think the use of the word "domain" in the context of GORAB localization at the *trans*-Golgi is correct. Nevertheless, we appreciate the referee's point about ambiguity, and have modified the text in the introduction (page 5), results (page 9), and discussion (page 20) to clarify what we mean by "domain".

It might be better to describe in more details regarding the glycan assignments in Figure 6.

We now include a description of how the glycan assignments were made in the legend to Figure 7 (previously Figure 6). The assignments are based on mass determination and well-established knowledge of the *N*-glycan biosynthetic pathway. We have added a supplementary table (Table S1) detailing the glycan species assigned, along with their mass and glycan type.

REVIEWERS' COMMENTS:

Reviewer #1 (Remarks to the Author):

The original submission was a good quality and interesting paper, but some additions were required to make the conclusions as watertight as possible. In the revised paper the authors have done an excellent job of addressing my concerns with both textual revisions and new and improved data. As such I am happy to recommend acceptance.

Reviewer #2 (Remarks to the Author):

The authors have addressed the concerns I raised.

The study shows how GORAB functions as a scaffolding protein at the TGN for Scyl1 and COPI to ensure proper recycling of Golgi enzymes.

Reviewer #3 (Remarks to the Author):

Most of changes and responses are satisfied.

Regarding the glycan assignments, nowadays with modern MS with high mass accuracy, we can get the mass with four digits after the decimal point, but in this work the masses of glycans are not highly accurate. Actually the mass is critically important for glycan assignments, and without accurate masses, the glycan assignments could be problematic.

It is well-known that normally glycans are hard to be ionized. Under the positive mode of MS, how are glycans positively charged? forming an adduct with metal ions or being protonated? This is also very important for glycan assignments. In the Table S1, what are the theoretical masses of these glycans? and what are the mass accuracies (ppm) between the experimental values and the corresponding theoretical ones? It might be better to list the theoretical mass and mass accuracy for each glycan in the Table.

Response to referee comments

Referee 3

Regarding the glycan assignments, nowadays with modern MS with high mass accuracy, we can get the mass with four digits after the decimal point, but in this work the masses of glycans are not highly accurate. Actually the mass is critically important for glycan assignments, and without accurate masses, the glycan assignments could be problematic.

It is well-known that normally glycans are hard to be ionized. Under the positive mode of MS, how are glycans positively charged? forming an adduct with metal ions or being protonated? This is also very important for glycan assignments. In the Table S1, what are the theoretical masses of these glycans? and what are the mass accuracies (ppm) between the experimental values and the corresponding theoretical ones? It might be better to list the theoretical mass and mass accuracy for each glycan in the Table.

Response: We have added more information to a new version of Table S1 to include the theoretical and actual measured masses of each of the indicated glycan species, to four decimal places. The mass accuracy for the each glycan is also indicated in the table. Regarding ionization of the glycans, they are positively charged predominantly through formation of sodium cationized species, as described in the references listed below. We are therefore confident in our glycan assignments.

Alkali metal, usually sodium, adducts are almost the universal products in the positive ion MALDI spectra of carbohydrates unless strongly basic groups are present, in which case some $[M + H]^+$ ions may be formed (Harvey DJ. Analysis of carbohydrates and glycoconjugates by matrix-assisted laser desorption/ionization mass spectrometry: An update covering the period 2001-2002. Mass Spectrom Rev. 2008 ; 27(2):125-201).

Ions produced by MALDI are generally singly charged molecular ions such as $[M+Na]^+$. (Jang-Lee J. et al. Glycomic profiling of cells and tissues by mass spectrometry: fingerprinting and sequencing methodologies. Methods Enzymol. 2006;415:59-86.)

Glycans yield intense signals corresponding to sodium cationized molecular species $[M+Na]^+$ in the positive ion mode.(Morelle et al., The use of mass spectrometry for the proteomic analysis of glycosylation. Proteomics 2006, 6, 3993–4015).